# Schur Nets: effectively exploiting local structure for equivariance in higher order graph neural networks

**Qingqi Zhang**[1*]**, Ruize Xu**[2*] **and Risi Kondor**[1,2,3]
[1]Computational and Applied Mathematics
Departments of [2]Computer Science and [3]Statistics
University of Chicago
{qingqi,richard1xur,risi}@uchicago.edu

## Abstract

Recent works have shown that extending the message passing paradigm to subgraphs communicating with other subgraphs, especially via higher order messages, can boost the expressivity of graph neural networks. In such architectures, to faithfully account for local structure such as cycles, the local operations must be equivariant to the automorphism group of the local environment. However, enumerating the automorphism groups of all subgraphs of interest and finding appropriate equivariant operations for each one of them separately is generally not feasible. In this paper we propose a solution to this problem based on spectral graph theory that bypasses having to determine the automorphism group entirely and constructs a basis for equivariant operations directly from the graph Laplacian. We show that this approach can boost the performance of GNNs on some standard benchmarks.

## 1 Introduction

Message pasing neural networks (MPNNs) are the most popular paradigm for building neural networks on graphs [18]. While MPNNs have proved to be remarkably effective in a range of domains from program analysis [1] to drug discovery [20], a series of both empirical [3] and theoretical [40, 10] results have shown that the fact that MPNNs are based on just vertices sending messages to their neighbors limits their expressive power. This problem is especially acute in domains such as chemistry, where the presence or absence of specific small structural units (functional groups) directly influences the behavior and properties of molecules.

Recent papers proposed to address this issue by extending the message paradigm to also allow for message passing between vertices and edges [24] or between subgraphs [2, 15, 4, 46]. However, if the actual messages remain scalars, the added expressive power of these networks is relatively limited.

A newer development is the appearance of *higher order MPNN*s, where the messages are not just scalars, but more complex objects indexed by the vertices of the sending and receiving subgraphs [31, 34, 14]. For example, in an organic molecule, the internal state of a benzene ring (six carbon atoms arranged in a cycle) can be represented as a matrix $T \in \mathbb{R}^{6 \times c}$, where the rows correspond to the individual carbons. When this benzene ring passes a message to a neighboring benzene ring that it shares an edge with, information relating to the two shared carbons can be sent directly to the corresponding atoms in the second ring. Information relating to the other vertices has to be treated differently. Hypergraph neural networks [15, 45, 41] and neural networks on more abstract mathematical structures such as simplicial complexes [7, 6] are closely related, since these also

---

* denotes equal contribution.

38th Conference on Neural Information Processing Systems (NeurIPS 2024).

involve higher order generalizations of message passing between combinatorial objects interlocked in potentially complicated ways.

In prior work we introduced a formalism called $P$-tensors that provides a flexible framework for implementing and reasoning about higher order MPNNs [21]. $P$-tensors build closely on the by now substantial literature on permutation equivariance in neural networks from Deep Sets [43], to various works studying higher order permutation equivariant maps [32]. It is also related to the category theoretic approach employed in Natural Graph Neural Networks [12].

One aspect of higher order message passing that has hithero not been studied in detail is the interaction between the local graph structure and the equivariance constraint. Intuitively, the reason that GNNs must be equivariant to permutations is that when the same, or analogous, graphs are presented to the network with the only difference being that the vertices have been renumbered, the final output must remain the same. However, Thiede et al. [37] observed that enforcing full permutation equivariance, especially at the local level, can be too restrictive. When considering specific, structurally important subgraphs such as paths or cycles, the operations local to such a subgraph $S$ should really only be equivariant to the automorphism group of $S$, rather than all $|S|!$ permutations. The Autobahn architecture described in [37] explicitly accounts for the automorpism group of two specific types of subgraphs, cycles and path. Defining bespoke convolution operations on these two types of subgraphs was shown to improve performance on molecular datasets like ZINC [26].

The drawback of the Autobahn approach is that it requires explicitly identifying the automorphism group of each type of subgraph, and crafting specialized equivariant operations based on its representation theory. For more flexible architectures leveraging not just cycles and paths but many other types of subgraphs this quickly becomes infeasible.

**Main contributions.** The main contribution of this paper is a simple algorithm based on spectral graph theory for constructing a basis of automorphism equivariant operations on any possible subgraph *without* explicitly having to determine its automorphism group, let alone derive its irreducible representations. The algorithm is based on imitating the structure of the group theoretical approach to equivariance, which we summarize in Section 3, but bypassing having to use actual group theory. Section 4 describes the algorithm itself and the resulting neural network operations, which we call *Schur layers*. In Section 5 we show that the introduction of Schur layers does indeed improve the performance of higher order graph neural networks on standard molecular benchmarks.

## 2 Background: equivariance with side information

Let $\mathcal{G}$ be an undirected graph with vertex set $V = \{1, 2, \ldots, n\}$ and edge set $E \subseteq V \times V$ represented by the adjacency matrix $A \in \mathbb{R}^{n \times n}$. Graph neural networks (GNNs) address one of two, closely related tasks: (a) given a function $f^{\text{in}} \colon V \to \mathbb{R}$ on the vertices of $\mathcal{G}$, learn to transform it to another function $f^{\text{out}} \colon V \to \mathbb{R}$; (b) given not just one graph, but an entire collection of graphs, learn to map each graph $\mathcal{G}$, or rather its adjacency matrix, to a scalar or vector $\psi(A)$ that characterizes $\mathcal{G}$'s structure.

In both cases, the critical constraint is that the network must behave appropriately under permuting the order in which the vertices are listed. The group of all permutations of $\{1, 2, \ldots, n\}$ is called the *symmetric group* of degree $n$ and denoted $\mathbb{S}_n$. Applying a permutation $\sigma \in \mathbb{S}_n$ to the vertices changes the adjacency matrix to $A' = \sigma(A)$ with

$$A'_{i,j} = A_{\sigma^{-1}(i), \sigma^{-1}(j)}. \tag{1}$$

The basic constraint on algorithms that operate on functions on graphs is that if the input is transformed along with the numbering of the vertices, $f_i^{\text{in}\prime} = f_{\sigma^{-1}(i)}^{\text{in}}$, then the output must transform the same way, $f_i^{\text{out}\prime} = f_{\sigma^{-1}(i)}^{\text{out}}$. This property is called *equivariance*. Formally, denoting the mapping from inputs to outputs $\phi \colon f^{\text{in}} \mapsto f^{\text{out}}$, equivariance states that the action $\sigma \colon f \mapsto f'$ of permutations on functions given by $f_i' = f_{\sigma^{-1}(i)}$ must commute with $\phi$, that is,

$$\phi(\sigma(f^{\text{in}})) = \sigma(\phi(f^{\text{in}})) \tag{2}$$

for any $\sigma \in \mathbb{S}_n$ and any input function $f^{\text{in}} \in \mathbb{R}^V$. In contrast, for networks that just learn embeddings $\mathcal{G} \mapsto \psi(A)$, the constraint is *invariance* to permutations, i.e., $\psi(A) = \psi(\sigma(A))$. In practice, the two cases are closely related because most graph embedding networks work with functions defined on the

vertices, and then symmetrize over permutations in their final layer using a readout function such as $\psi(\mathcal{G}) = \sum_i f_i^{\text{out}}$.

Enforcing (2) on the hypothesis space directly would be very limiting, effectively constraining GNNs to be composed of symmetric polynomials of $f^{\text{in}}$ combined with pointwise nonlinearities. Message passing graph neural networks cleverly get around this problem by using the adjacency matrix itself as *side information* to guide equivariance. For example, in classical (zeroth order) MPNNs, the output of each layer is a (vector valued) function on the graph, and the update rule from layer $\ell$ to layer $\ell + 1$ in the simplest case is

$$f_i^{\ell+1} = \eta\Big(W \sum_{j \in \mathcal{N}(i)} f_j^\ell + \theta\Big), \tag{3}$$

where $\mathcal{N}(i)$ denotes the neighbors of node $i$ in $\mathcal{G}$, $W$ and $\theta$ are learnable weight matrices/vectors and $\eta$ is a suitable nonlinearity. The fact that the summation only extends over the neighbors of vertex $i$ induces a dependence of $\phi$ on the adjacency matrix $A$. Hence it would be more accurate to use the notation $\phi_A$ for the mapping, and write the equivariance condition as

$$\phi_{\sigma(A)}(\sigma(f^{\text{in}})) = \sigma(\phi_A(f^{\text{in}})). \tag{4}$$

In this interpretation, when the vertices are permuted, $A$ can implicitly provide information about this to the algorithm, allowing it to compensate. However, $A$ cannot convey any information about permutations that leave it invariant, so $\phi_A$ must still be equivariant to the group of all such permutations, called the *automorphism group* of $\mathcal{G}$, denoted $\text{Aut}(\mathcal{G})$ or just $\text{Aut}(A)$. This observation about $A$ acting as a source of side information that can reduce the size of the group that individual GNN operations need to be equivariant to is the starting point for the rest of this paper.

## 2.1 Higher order GNNs

Despite the great success of message passing networks, a long sequence of empirical as well as theoretical results [40, 34, 30, 13, 35] over the last several years have made it clear that the expressive power of algorithms based on simple update rules like (3) is severely limited. In response, the community has extended the message passing paradigm to message passing between vertices and edges or between carefully selected subgraphs [18, 34, 6, 8]. These networks maintain the local and equivariant character of earlier MPNNs, but they can more faithfully reflect local topological information and are particularly well suited to domains such as chemistry, where capturing local structures such as functional groups is critical.

In tandem, researchers have developed *higher order MPNN* architectures, where the outputs of individual edge or subgraph "neurons" are not just scalars, but vector or tensor quantities indexed by the vertices of the graph involved in the given substructure. For example, in the chemistry setting, if $\mathfrak{n}$ is a titular neuron corresponding to a benzene ring in a molecule, the output of $\mathfrak{n}$, assuming we have $C$ channels, might be a matrix $T \in \mathbb{R}^{6 \times C}$ or a tensor $T \in \mathbb{R}^{6 \times 6 \times C}$, or $T \in \mathbb{R}^{6 \times 6 \times 6 \times C}$, etc.. In each of these cases the non-channel dimensions correspond to the six atoms making up the ring, and when $\mathfrak{n}$ communicates with other subgraph-neurons, must be treated accordingly. Such higher order representations offer greater expressive power because they allow $T$ to capture information about *relations* between pairs of vertices, triples, and so on. The general trend towards studying higher order message passing is also closely tied to the emergence of hypergraph neural networks [15], "topological" neural networks and simplicial complex networks [7, 6].

Recently, [21] proposed a general formalism for describing such higher order architectures using so-called $P$-tensors. In this formalism, given a neuron $\mathfrak{n}$ attached to a subgraph $S$ with $m$ vertices, we say that the output of $\mathfrak{n}$ is a *zeroth order $P$-tensor* $T$ with $C$ channels if it is simply a vector $T \in \mathbb{R}^C$. The elements of this vector are scalars in the sense that they are invariant to permutations of the vertices of $S$. We say that $T$ is a *first order $P$-tensor* if it is a matrix $T \in \mathbb{R}^{m \times C}$ whose columns transform under permutations of $S$ similarly to how $f^{\text{in}}$ and $f^{\text{out}}$ transform under global permutations of the full graph:

$$\sigma : T \mapsto T' \qquad\qquad T'_{i,c} = T_{\sigma^{-1}(i),c} \qquad\qquad \sigma \in \mathbb{S}_m. \tag{5}$$

We say that $T$ is a *second order $P$-tensor* $T \in \mathbb{R}^{m \times m \times C}$, if each slice corresponding to a given channel transforms according to the second order action of the symmetric group, similarly (1):

$$\sigma : T \mapsto T' \qquad\qquad T'_{i,j,c} = T_{\sigma^{-1}(i),\,\sigma^{-1}(j),c} \qquad\qquad \sigma \in \mathbb{S}_m. \tag{6}$$

Continuing this pattern, a *k'th order P-tensor* $T \in \mathbb{R}^{m \times m \times \dots \times m \times C}$ transforms under local permutations as

$$\sigma \colon T \mapsto T' \qquad\qquad T'_{i_1, \dots, i_k, c} = T_{\sigma^{-1}(i_1), \dots, \sigma^{-1}(i_k), c} \qquad\qquad \sigma \in \mathbb{S}_m. \qquad (7)$$

[21] derives the general rules for equivariant message passing between such $P$-tensors in the cases that the sending and receiving subgraphs $S$ resp. $S'$ are (a) the same (b) partially overlap (c) are disjoint. However, in each of these cases however it was assumed that $T$ needs to be equivariant to *all possible* permutations of the vertices of the underlying subgraphs $S$ and $S'$.

As we discussed above, this is an overly restrictive condition that limits the extent to which a higher order subgraph neural network can exploit the underlying topology. In the following sections we focus on just the type of messages that are sent from a given subgraph $S$ to *itself* (called *linmaps* in the $P$-tensors nomenclature), and derive a way of making these messages equivariant to just $\mathrm{Aut}(S)$ rather than the full symmetric group $\mathbb{S}_m$.

## 3 Equivariance to local permutations: the group theoretic approach

Recall that a *representation* of a finite group $G$ such as $\mathbb{S}_m$ or $\mathrm{Aut}(S)$ is a (complex) matrix valued function $\rho \colon G \to \mathbb{C}^{d_\rho \times d_\rho}$ where the $\rho(\sigma)$ matrices multiply the same way as the corresponding group elements do

$$\rho(\sigma_2 \sigma_1) = \rho(\sigma_2) \rho(\sigma_1) \qquad\qquad \sigma_1, \sigma_2 \in G.$$

Two representations $\rho$ and $\rho'$ are said to be *equivalent* if there is an invertible matrix $Q$ such that $\rho'(\sigma) = Q^{-1} \rho(\sigma) Q$ for all group elements. A representation of a finite group is said to be *reducible* if there is some invertible matrix $Q$ that reduces it to a block diagonal form

$$\rho(\sigma) = Q^{-1} \left( \begin{array}{c|c} \rho_1(\sigma) & \\ \hline & \rho_2(\sigma) \end{array} \right) Q.$$

Some fundamental results in representation theory tell us that G only has a finite number of inequivalent irreducible representations (*irreps*, for short), these irreps can be chosen to all be unitary, and that any representation of $G$ is reducible to some combination of them [36]. These facts give rise to a type of generalized Fourier analysis on finite groups that can decompose vectors that $G$ acts on into parts transforming according to the unitary irreps of the group.

The general approach to defining $\mathrm{Aut}(S)$-equivariant maps for first, second, and higher order subgraph neurons would use this machinery. In particular, defining permutation matrices as usual as

$$[P_\sigma]_{i,j} = \begin{cases} 1 & \text{if } \sigma(j) = i \\ 0 & \text{otherwise,} \end{cases}$$

dropping the channel indices without loss of generality, and writing our $P$-tensors in vectorized form $\overline{T} = \mathrm{vec}(T) \in \mathbb{R}^{m^k}$, (5)–(7) can be written in a unified form

$$\overline{T'} = P^k(\sigma) \overline{T}$$

where $P^k(\sigma)$ is the $k$-fold Kronecker product matrix $P^k(\sigma) = P_\sigma \otimes P_\sigma \otimes \dots P_\sigma$. Crucially, as $\sigma$ ranges over the automorphisms of $S$, these product matrices $P^k(\sigma)$, form a unitary representation of the automorphism group.

According to representation theory, $P^k$ must then be decomposable into a direct sum of irreps $\rho_1, \dots, \rho_p$ of $\mathrm{Aut}(S)$ with corresponding multiplicities $\kappa_1, \dots, \kappa_p$. The same unitary matrix $Q$ that accomplishes this can also be used to decompose $\overline{T}$ into a combinations of smaller vectors $(\boldsymbol{t}_j^i \in \mathbb{R}^{d_{\rho_i}})_{i,j}$:

$$Q\overline{T} = \bigoplus_i \bigoplus_{j=1}^{\kappa_i} \boldsymbol{t}_j^i,$$

where each $\boldsymbol{t}_j^i$ now transforms independently under the action of the group as $\boldsymbol{t}_j^i \mapsto \rho_i(\sigma) \, \boldsymbol{t}_j^i$. Alternatively, stacking all $\boldsymbol{t}_j^i$ vectors transforming according to the same irrep $\rho_i$ together in a matrix $\boldsymbol{T}^i \in \mathbb{R}^{d_{\rho_i} \times \kappa_i}$, we arrive at a sequence of matrices $\boldsymbol{T}^1, \boldsymbol{T}^2, \dots, \boldsymbol{T}^p$ transforming as

$$\boldsymbol{T}^1 \mapsto \rho_1(\sigma) \, \boldsymbol{T}^1 \qquad\quad \boldsymbol{T}^2 \mapsto \rho_2(\sigma) \, \boldsymbol{T}^2 \qquad\quad \dots \qquad\quad \boldsymbol{T}^p \mapsto \rho_p(\sigma) \, \boldsymbol{T}^p. \qquad (8)$$

It is very easy to see how one might construct learnable linear operations that are equivariant to these actions: simply multiply each $\boldsymbol{T}^i$ *from the right* by a learnable weight matrix $W^i$.

This general, group theoretic approach to constructing automorphism group equivariant linear maps between $k$'th order $P$-tensors can be seen as a special case of [29, 38]. Operationally, it just reduces to the following sequence of steps:

1. Find the unitary matrix $Q$ that decomposes $P^k(\sigma) = P_\sigma \otimes \ldots \otimes P_\sigma$ into a direct sum of $\mathrm{Aut}(S)$ irreps.
2. Use $Q$ to decompose the input $P$-tensor into a sequence of matrices $\boldsymbol{T}^1 \ldots \boldsymbol{T}^p$ transforming as (8).
3. Multiply each $\boldsymbol{T}^i$ by an appopriate learnable weight matrix $W^i \in \mathbb{R}^{\kappa_i \times \kappa_i}$.
4. Use the inverse map $Q^{-1} = Q^\dagger$ to reassemble $\boldsymbol{T}^1, \boldsymbol{T}^2, \ldots, \boldsymbol{T}^p$ into the output $P$-tensor $T^{\mathrm{out}}$.

Notwithstanding its elegance, this representation approach to implementing automorphism group equivariance also has some disadvantages. Specifically, it requires to (a) determine the automorphism group of each subgraph, and (b) explicitly find its irreducible representations, which is also not trivial. The underlying mathematical structure however is important because it forms the basis to generalizing the approach to a much simpler framework in the next section:

1. We have a collection of (orthogonal) linear maps $\{P^k(\sigma)\colon U \mapsto U\}_{\sigma \in \mathrm{Aut}(S)}$ (with $U = \mathbb{R}^{m^k}$) that the neuron's operation needs to be equivariant to.
2. $U$ is decomposed into an orthogonal sum of subspaces $U = U_1 \oplus \ldots \oplus U_p$ corresponding to the different irreps featured in the decomposition of $P^k$.
3. Each $U_i$ is further decomposed into an orthogonal sum of subspaces $U_i = V_1^i \oplus \ldots \oplus V_{\kappa_i}^i$ corresponding to the different columns of the $\boldsymbol{T}^i$ matrices.
4. The decomposition is such that the $\{P^k(\sigma)\}$ maps fix each $V_j^i$ subspace. Moreover, for a fixed $i$, $\{P^k(\sigma)\}$ acts the *same* way on each $V_j^i$ subspace by $\rho_i(\sigma)$.
5. This structure implies that any linear map that linearly mixes the $V_1^i, \ldots V_{\kappa_i}^i$ subspaces but does *not* mix information across subspaces with different values of $i$ is equivariant.

## 4 Equivariance via spectral graph theory: Schur layers

In place of the representation theoretical approach described in the previous section, in this paper we advocate a simpler way of implementing automorphism group equivariance based on just spectral graph theory. The cornerstones of this approach are the following two theorems. The proofs can be found in the Appendix.

**Theorem 1.** *Let $G$ be a finite group acting on a space $U$ by the linear action $\{g\colon U \to U\}_{g \in G}$. Assume that we have a decomposition of $U$ into a sequence of spaces of the form*

$$U = U_1 \oplus \ldots \oplus U_p$$

*where each $U_i$ is invariant under the action of the group (this means that for any $g \in G$ and $v \in U_i$, we have $g(v) \in U_i$). Let $\phi\colon U \to U$ be a linear map that is a homothety on each $U_i$, i.e., $\phi(w) = \alpha_i w$ for some fixed scalar $\alpha_i$ for any $w \in U_i$. Then $\phi$ is equivariant to the action of $G$, i.e., $\phi(g(u)) = g(\phi(u))$ for any $u \in U$ and any $g \in G$.*

The representation theoretic result of the previous section corresponds to a refinement of this result involving a further decomposition of each $U_i$ space into a sequence of smaller subspaces.

**Theorem 2.** *Let $G$ and $U$ be as in Theorem 1, but now assume that each $U_i$ further decomposes into an orthogonal sum of subspaces in the form $U_i = V_1^i \oplus \ldots \oplus V_{\kappa_i}^i$ such that*
*(a) Each $V_j^i$ subspace is individually invariant by $G$;*
*(b) For any fixed value of $i$, the spaces $V_1^i, \ldots, V_{\kappa_i}^i$ are isomorphic and there is a set of canonical isomorphisms $\iota_{j \to j'}^i\colon V_j^i \to V_{j'}^i$ between them such that*

$$g(\iota_{j \to j'}^i(v)) = \iota_{j \to j'}^i(g(v)) \qquad\qquad \forall v \in V_j^i.$$

*Let $\phi\colon U \to U$ be a map of the form*

$$\phi(v) = \sum_{j'} \alpha_{j,j'}^i \, \iota_{j \to j'}^i(v) \qquad\qquad v \in V_j^i$$

*for some fixed set of coefficients $\{\alpha_{j,j'}^i\}$. Then $\phi$ is equivariant to the action of $G$ on $U$.*

In the matrix language of the previous section, $U_1, \ldots, U_p$ correspond to the $\boldsymbol{T}^1, \boldsymbol{T}^2, \ldots, \boldsymbol{T}^p$ matrices, whereas the $V_1^i, \ldots, V_{\kappa_i}^i$ subspaces of $U_i$ correspond to individual columns of $\boldsymbol{T}^i$. For any fixed $i$, the $\left(\alpha_{j,j'}^i\right)_{j,j'}$ scalars correspond to the individual matrix entries of the learnable weight matrix $W^i$. For our simplified spectral approach to automorphism group equivariance we will content ourselves with using Theorem 1 rather than Theorem 2.

### 4.1 Automorphism invariance the simple way via spectral graph theory

The key insight of this paper is that we do not necessarily need to use heavy representation theoretic machinery to find a system of subspaces to plug into Theorem 1. In particular, we have the following simple lemma.

**Lemma 1.** *Let $S$ be an undirected graph with $m$ vertices, $\mathrm{Aut}_S$ its automorphism group, and $L$ its combinatorial graph Laplacian. Assume that $L$ has $p$ distinct eigenvalues $\lambda_1, \ldots, \lambda_p$ and corresponding subspaces $U_1, \ldots, U_p$. Then each $U_i$ is invariant under the first order action (5) of $\mathrm{Aut}_S$ on $\mathbb{R}^m$.*

*Proof.* Since $\mathrm{Aut}_S$ is a subgroup of the full group of vertex permutations , its action on $\mathbb{R}^m$ is just $\mathbf{v} \mapsto \sigma(\mathbf{v}) = P_\sigma \mathbf{v}$ with $\sigma \in \mathrm{Aut}_S$. $L$ is a real symmetric matrix, so its eigenspaces $U_1, \ldots, U_p$ are mutually orthogonal and $U_1 \oplus \ldots \oplus U_p = \mathbb{R}^n$. Furthermore, $\mathbf{v} \in U_i$ if and only if $L\mathbf{v} = \lambda_i \mathbf{v}$. By definition, $\mathrm{Aut}_S$ is the set of permutations that leave the adjacency matrix, and consequently the Laplacian invariant, so. In particular, $P_\sigma L P_{\sigma^{-1}} = L$ for any $\sigma \in \mathrm{Aut}_S$. Therefore, for any $\mathbf{v} \in U_i$

$$L\left(\sigma(\mathbf{v})\right) = L P_\sigma \mathbf{v} = P_\sigma L P_{\sigma^{-1}} P_\sigma \mathbf{v} = P_\sigma L \mathbf{v} = \lambda_i P_\sigma \mathbf{v} = \lambda_i \sigma(\mathbf{v}) \qquad \forall\, \sigma \in \mathrm{Aut}_S$$

showing that $\sigma(\mathbf{v}) \in U_i$. Hence $U_i$ is an invariant subspace. $\qquad\square$

The following Corollary puts this lemma to use, providing a surprisingly easy way of creating locally automorphism equivariant neurons. We define a *Schur layer* as a neural network module that applies this operation to every instance of a given subgraph in the graph, for example, every benzene ring in a molecule. For the sake of global permutation equivariance, the weight matrices for any given subgraph $S$ must be shared across all instances of $S$ across in the graph.

**Corollary 1.** *Consider a GNN on a graph that involves a neuron $\mathfrak{n}_S$ corresponding to a subgraph $S$ with $m$ vertices. Assume that the input of $\mathfrak{n}_S$ is a matrix $T \in \mathbb{R}^{m \times c_{in}}$, the rows of which transform covariantly with permutations of $S$ and $c_{in}$ is the number of channels. Let $L$ be the combinatorial Laplacian of $S$, $U_1, \ldots, U_p$ be the eigenspaces of $L$, and $M_i$ an orthognal basis for the $i$'th eigenspace stacked into an $\mathbb{R}^{n \times dim(U_i)}$ dimensional matrix. Then for any collection of learnable weight matrices $W_1, \ldots, W_p \in \mathbb{R}^{c_{in} \times c_{out}}$,*

$$\phi \colon T \longmapsto \sum_{i=1}^{p} M_i M_i^\top T W_i \tag{9}$$

*is a permutation equivariant linear operation.*

The spectral approach also generalizes to higher order permutation equivariance, in which case we can take advantage of the more refined two-level subspace structure implied by Theorem 2.

**Theorem 3.** *Let $S$, $L$ and the $M_i$'s be as in Corollary 1. Given a multi-index $\mathbf{i} = (i_1, \ldots, i_k) \in \{1, \ldots, p\}^k$, we define its type as the tuple $\mathbf{n} = (n_1, \ldots, n_p)$, where $n_j$ is the number of occurrences of $j$ in $\mathbf{i}$ and we define $\mathcal{I}_\mathbf{n}$ as the set of all multi-indices of type $\mathbf{n}$. Assume that the input to neuron $\mathfrak{n}_S$ is a $k$'th order permutation equivariant tensor $T \in \mathbb{R}^{m \times \ldots \times m \times c_{in}}$, as defined in Section 3. For any given $\mathbf{i}$, define the $k$'th order eigen-projector*

$$\Pi_\mathbf{i} = \mathcal{P}_\mathbf{i}(M_{i_1}^\top \otimes M_{i_2}^\top \otimes \ldots \otimes M_{i_k}^\top \otimes I) \colon \mathbb{R}^{m \times \ldots \times m \times c} \to \mathbb{R}^{m \times \ldots \times m \times c}$$

*where $\mathcal{P}_\mathbf{i}$ is a permutation map that canonicalizes the form of the projection, as defined in the Appendix. Let $T \circ W$ denote multiplying $T$ by the matrix $W$ only along its last dimension. Then for any collection of weight matrices $\{W_{\mathbf{i}',\mathbf{i}'} \in \mathbb{R}^{c_{out} \times c_{in}}\}$ the map*

$$\phi \colon T \longmapsto \sum_\mathbf{n} \sum_{\mathbf{i}' \in \mathcal{I}_\mathbf{n}} \sum_{\mathbf{i} \in \mathcal{I}_\mathbf{n}} \Pi_{\mathbf{i}'}^\top (\Pi_\mathbf{i}(T \odot W_{\mathbf{i}',\mathbf{i}'}))$$

*is a permutation equivariant map.*

Corollary 1 and Theorem 3 give sufficient but not necessary conditions for equivariance w.r.t. the automorphism group. A discussion of the potential gap between our approach and the full group theoretical approach described in Section 3 can be found in Appendix D.

# 5 Experiments

To empirically evaluate our Schur layers, we implement the first order case described in Corollary 1 and compare it with other higher order MPNNs. Note that the theorem only gives the equivariant maps to transform local representation on a subgraph, i.e., $\phi : T^{\text{in}} \mapsto T^{\text{out}}$ where $T^{\text{in}} \in \mathbb{R}^{m \times c_{\text{in}}}$ and $T^{\text{out}} \in \mathbb{R}^{m \times c_{\text{out}}}$ are the inputs and output of a neuron corresponding to a specific subgraph. The rest of the message passing is conducted in the usual way, in particular, everything is implemented in the $P$-tensors framework [21].

There are several design details about the architecture that are worth mentioning: (1) We chose cycles of lengths three to eight in most of the experiments and also added branched cycles to show our algorithm's scalability. The reason is that in the chemical dataset we used, cycles are the most important functional group to the property to be predicted, other subgraphs such as $m$-stars, and $m$-paths did not help the performance. (2) While having first-order representation on the cycles, we also maintain 0'th-order representation (i.e., scalars) on node and edges, as in [6, 21]. These node and edge representations capture more elementary information about the graph, such as $k$-hop neighborhoods, and are still important in higher order MPNNs. The representations pass messages with each other by intersection rule as defined in $P$-tensor framework. (3) As discussed in Appendix G, we view *Schur* layer's operation as a spectral convolution filter [5] applied to the subgraph and the number of channels to indicate how many times it expands the input feature to the output feature. The codes used to run our experiments can be found at https://github.com/risilab/SchurNet.

## 5.1 *Schur* layer improves over *Linmaps*

First of all, we want to show that considering more possible equivariant maps on the subgraph can indeed boost the performance. To this end, we performed controlled experiments to compare *Schur* layer with the equivariant maps w.r.t. $\mathbb{S}_m$ (following [21] we called these *linmaps*). To make a fair comparison, we use the same architecture including MLPs and message passing defined by $P$-tensor and only replace the equivariant maps used in *Linmaps* by what is defined in Corollary 1. We didn't compare with Autobahn [37] because it didn't use the $P$-tensor framework in the original paper and it's hard for us to implement the convolution w.r.t. automorphism group for all the subgraphs we chose. However we note that for the case of the cycles (see Table 4), the equivariant maps given by our approach are equal to that given by the group theoretical approach.

We'll present the results on the commonly used molecular benchmark ZINC-12K [26] dataset in the main text. Results on TUdatasets as well as runtime comparison can be found in Appendix H. The task on ZINC-12K is to regress the $\log P$ coefficient of each molecule and the Mean absolute error (MAE) between the prediction and the ground truth is used for evaluation.

We design experiments to compare *Linmaps* and *Schur* layers in various scenarios, to showcase the robustness of the improvement. Table 1 shows under various message passing schemes between edges and cycles, *Schur* Layer consistently outperforms *Linmaps*. Those are indications of the added expressive power of extra equivariant maps in *Schur* layer, and they're effective in various architectural design settings. Another ablation study regarding different cycle sizes can be found in Appendix H.

| Layer | Pass message when overlap $\geq 1$ | Pass message when overlap $\geq 2$ |
|:---:|:---:|:---:|
| *Linmaps* (baseline) | $0.074 \pm 0.008$ | $0.074 \pm 0.005$ |
| *Schur* layer | $\mathbf{0.070 \pm 0.006}$ | $\mathbf{0.071 \pm 0.003}$ |

Table 1: Comparison between *Schur* Layer and *Linmaps* with different message passing schemes. The message passing scheme is a design choice in $P$-tensor framework, where the user can set when two subgraph's representations communicate. The mostly common use case is to require at least $k$ vertices in the intersection of two subgraphs for them to communicate. Experiments on ZINC-12k dataset and all scores are test MAE. Cycle sizes of {3,4,5,6,7,8,11} are used.

Then we studied the possibilities of adding *Schur* layer in different places of higher-order message passing scheme. In table 2, we observed that the more condensed the higher-order feature is, the more improvement that the *Schur* Layer brings to us over *Linmaps*. We attribute the improvements of adding/replacing *Schur* layer in various scenarios over *Linmaps* the benefit gained from utilizing the

subgraph structure and increased number of equivariant maps. We also tried other ways to use *Schur* layer, the result is summarized in Appendix G.

| Model | Test MAE |
|---|---|
| *Linmaps* | $0.071 \pm 0.004$ |
| Simple *Schur*-Net | $0.070 \pm 0.005$ |
| Linmap *Schur*-Net | $\mathbf{0.068 \pm 0.002}$ |
| Complete *Schur*-Net | $\mathbf{0.064 \pm 0.002}$ |

Table 2: An experiment demonstrating different ways of using *Schur* layer. "Complete *Schur* Layer" means that we apply *Schur* Layer on the incoming messages together with the original cycle representation. "Linmap SchuLayer" means that we just apply the *Schur* Layer on the aggregated subgraph representation feature. "Simple *Schur* Layer" means we directly apply *Schur* Layer on the subgraph features without any preprocessing. We can observe that as the subgraph information diversifies, *Schur* layer tends to decouple the dense information better and results in better performance. The test MAE of *Linmaps* in this table is taken from [21].

## 5.2 Flexibility

The other advantage of *Schur* layer is that it computes the feature transform only based on the subgraph's Laplacian, bypassing a difficult step of finding the automorphisms group and *irreps* of the subgraph it acts on. As discussed in the theory, *Schur* layer constructs equivariant maps only based on the subgraph's Laplacian and is applicable directly to any subgraphs, making the implementation much easier when different subgraphs are chosen than the group theoretical approach. This allows it to easily extend to any subgraph templates that're favorable by the user. To demonstrate this, we augment the subgraphs in the model by all the five and six cycles with one to three branches (including in total 16 non-isomorphic subgraph templates), comparing with baseline model where only the cycle itself is considered. Results can be found in Appendix G.

## 5.3 Benchmark results

Finally, and most importantly, we compare the *Schur*-Net to several other higher-order MPNNs [1] on ZINC-12k dataset and OGB-HIV dataset [23] in table 3. We included baselines of (1) classical MPNNs: GCN[28], GIN [40], GINE [24], PNA[11], HIMP [16] (2) higher order MPNNs: $N^2$-GNN [14] [2], CIN [6], *P*-tensors [21] (3) Subgraph-GNNs: DS-GNN(EGO+) and DSS-GNN(EGO+) [4], GNN-AK+ [46], SUN(EGO+)[17] (4) Autobahn [37].

We find that *Schur* Net ranked second on ZINC-12K and outperformed all other baselines on OGB-HIV dataset. This shows the expressivity of adding more equivariant maps by leveraging the subgraph topology. Furthermore, note that while $N^2$-GNN outperforms *Schur* Net on ZINC-12K, it's a second-order model whereas in our experiment, we only used first-order activation. Also, the partial reason *Autobahn* didn't perform well is in the original implementation, the authors didn't use *P*-tensor framework and used only a part of all possible linear message passing schemes. This shows to get the full power of equivariant maps w.r.t. subgraph automorphism group, we need to combine it with a general message passing framework between subgraphs as well.

# 6 Limitations

Unlike the representation theoretic approach, the spectral approach is not guaranteed the give the finest possible decomposition into invariant subspaces. Hence, equations like (9) do not necessarily define the most general possible automorphism-equivariant linear maps. In this paper we did not investigate from a theoretical point of view the extent of this gap. In general, being able to craft automorphism-equivariant layers of any order for any types of subgraphs opens up a host of possibilities for making GNNs more powerful. Our experiments are limited to some of the simplest cases, such as exploiting cycles and edges. We also only used first order activations.

---

[1] A discussion of related work can be found in Appendix C.
[2] The works [31, 34] don't have results in those two dataset.

| Model | ZINC-12K MAE($\downarrow$) | OGB-HIV ROC-AUC($\% \uparrow$) |
|---|---|---|
| GCN | $0.321 \pm 0.009$ | $76.07 \pm 0.97$ |
| GIN | $0.408 \pm 0.008$ | $75.58 \pm 1.40$ |
| GINE | $0.252 \pm 0.014$ | $75.58 \pm 1.40$ |
| PNA | $0.133 \pm 0.011$ | $79.05 \pm 1.32$ |
| HIMP | $0.151 \pm 0.002$ | $78.80 \pm 0.82$ |
| $N^2$-GNN | $\mathbf{0.059 \pm 0.002}$ | - |
| CIN | $0.079 \pm 0.006$ | $80.94 \pm 0.57$ |
| P-tensors | $0.071 \pm 0.004$ | $80.76 \pm 0.82$ |
| DS-GNN (EGO+) | $0.105 \pm 0.003$ | $77.40 \pm 2.19$ |
| DSS-GNN (EGO+) | $0.097 \pm 0.006$ | $76.78 \pm 1.66$ |
| GNN-AK+ | $0.091 \pm 0.011$ | $79.61 \pm 1.19$ |
| SUN (EGO+) | $0.084 \pm 0.002$ | $80.03 \pm 0.55$ |
| Autobahn | $0.106 \pm 0.004$ | $78.0 \pm 0.30$ |
| *Schur*-Net | $0.064 \pm 0.002$ | $\mathbf{81.6 \pm 0.295}$ |

Table 3: Comparison of different models on the ZINC-12K and OGBG-MOLHIV datasets.

## 7    Conclusions

Enforcing equivariance to the automorphism group of subgraphs in higher order neural networks seemingly requires the use of advanced tools from group representation theory. This is likely a large part of the reason why automorphism-based architectures such as [37] are not used more commonly in practical applications. In this paper we have shown that a simpler approach based on spectral graph theory, following the same underlying logic as the group theoretic approach but bypassing having to enumerate all irreducible representations of the automorphism group, can lead to an architecture that is almost as expressive. Our algorithm, called Schur Nets, easily generalizes to higher order activations, as well as incorporating other types of side information such as vertex labels. In a practical setting, Schur Nets is easiest to deploy in conjunction with a message passing framework like $P$-tensors that hides the complexities of the higher order message passing component. The empirical performance of Schur Nets on the ZINC 12K dataset is superiror to all other comparable (non-transformer based) architectures that we are aware of.

Given the similarity between the way we utilize the eigenspaces of the graph Laplacian and the so-called graph Fourier transform, our approach exposes heretofore unexplored connections between permutation equivariance and spectral GNNs such as [9, 22]. It also highlights the fact that while permutation equivariance is a fundamental constraint on graph neural networks, the key to building high performing, expressive GNNs is to reduce the size of the group that the network needs to be equivariant to as much as possible, using whatever side-information we can employ, whether that be the adjacency matrix, vertex degrees or something else.

## Acknowledgements

We would like to thank Andrew Hands for many valuable pieces of advice and much practical assistance that he has given to the experimental side of this project. We also gratefully acknowledge the Toyota Technological Institute of Chicago and the Data Science Institute at the University of Chicago for making their computational resources available for our use, as well as NSF MRI-1828629 for additional infrastructure that was used in the course in this project.

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

# A   Additional details

The purpose of the permutation map $\mathcal{P}_\mathbf{i}$ in Theorem 3 is to remap the tensor product space $\mathbb{R}^{\dim(U_{i_1}) \times \ldots \times \dim(U_{i_k}) \times C}$ into a canonical form so that those dimensions that correspond to $i_j = 1$ are mapped to the first $n_1$ slots, those dimensions with $i_j = 2$ are mapped to the second $n_2$ slots, and so on. Given the permutation $\tau \colon \{1, 2, \ldots, k\} \to \{1, 2, \ldots, k\}$ that canonicalizes the indices in this way, $\mathcal{P}_\mathbf{i}$ is just the corresponding permutation map, i.e.,

$$\mathcal{P}_\mathbf{i}(x_1 \otimes \ldots x_k \otimes x_c) = x_{\tau(1)} \otimes \ldots x_{\tau(k)} \otimes x_c.$$

# B   Proofs

**Proof of Theorem 1.** Let $\{g\!\downarrow_{U_i} \colon U_i \to U_i\}_{g \in G}$ be the restriction of the action of $G$ to $U_i$. Since $G$ fixes each $U_i$, the action of $G$ on the full space decomposes in the form

$$g(u) = \sum_{i=1}^{p} g\!\downarrow_{u_i}(u\!\downarrow_{U_i}) \qquad\qquad u \in U.$$

Now since multiplication by scalars commutes with linear maps,

$$\phi(g(u)) = \sum_{i=1}^{p} \alpha_i \, g\!\downarrow_{u_i}(u\!\downarrow_{U_i}) = \sum_{i=1}^{p} g\!\downarrow_{u_i}(\alpha_i \, u\!\downarrow_{U_i}) = g(\phi(u))$$

for any $u \in U$ and any $g \in G$. $\qquad\blacksquare$

**Proof of Theorem 2.** Each of the $V_j^i$ subspaces is invariant, hence a homothety on each is an equivariant operation for the same reason as in Theorem 1. In addition, since $G$ acts the same way on any pair of subspaces $(V_j^i, V_{j'}^i)$, any map of the form $\xi_{j,j'}^i \colon V_j^i V_{j'}^i$ that is just a scaling is also equivariant. The composition of equivariant linear maps is an equivariant linear map, hence any map $\phi$ of the given form is equivariant. $\qquad\blacksquare$

**Proof of Theorem 3.** For any multi-index $\mathbf{i}$, the subspace $\mathrm{Im}(M_{i_1}^\top \otimes M_{i_2}^\top \otimes \ldots \otimes M_{i_k}^\top \otimes I_\mathrm{c})$ is invariant to permutations. Further after applying the permutation map $\mathcal{P}_\mathbf{i}$, all such subspaces of the same type $\mathbf{n}$, the action of $\mathbb{S}_k$ on all such subspaces will be the same. Hence we can directly apply Theorem 2 to prove this theorem. $\qquad\blacksquare$

## B.1   Alternative Proof for Corollary 1

Here we give a more direct proof for Corollary 1.

*Proof.* Without loss of generality, we assume $c_{in} = c_{out} = 1$. Since the $M_i$'s and $\lambda_i$'s are eigenspaces and eigenvectors of the Laplacian $L$, $L = M\Lambda M^T$, where $M = [M_1, \cdots, M_p]$ and $\Lambda = \mathrm{diag}(\lambda_1, \cdots, \lambda_m)$.

The neuron $\mathfrak{n}_S$ is a linear transform $\phi : R^m \to R^m$, with $\phi(T) = \Sigma_i^p M_i M_i^T T W_i = MVM^T T$, where $V = \mathrm{diag}(W_1 I_{n_1}, \cdots, W_p I_{n_p})$. Here the $W_i$'s are just scalars and $I_{n_i}$ is an identity matrix of the same size as the corresponding eigenspace.

Given a permutation $\sigma \in S_n$, we want to show that $\phi_\sigma(\sigma \circ T) = \sigma \circ \phi_e(T)$, which is the definition of equivariance. Note that $\phi_\sigma$ actually depends on $\sigma$, since we're doing eigenvalue decomposition on the Laplacian of the subgraph transformed by $\sigma$. This is the key to proving equivariance. We use $e$ to denote the identity permutation.

Now $L^\sigma = P_\sigma L P_\sigma^T = M_\sigma \Lambda M_\sigma^T$, where $M_\sigma = P_\sigma M$, while the eigenvalues remain invariant after permutation and eigenspaces are transformed in the same manner as $L$. Thus,

$$\begin{aligned}
\phi_\sigma(\sigma \circ T) &= M_\sigma V M_\sigma^\top P_\sigma T \\
&= P_\sigma M V M_\sigma^\top P_\sigma^\top P_\sigma T \\
&= P_\sigma \, \phi_e(T)
\end{aligned}$$

which is the desired result.

□

**Remark 1.** *The key to the proof is that the transform $\phi$ depends on an object transformed with permutation $\sigma$, specifically. graph Laplacian L. And all we need is that the object L is transformed equivariantly with $\sigma$. We can also add side informatino such as node labels or degrees to further constrain the automorphism group and increase the number of distinct eigenvalues. One easy way to do that is to set $L' = L + V$, where $V = diag(v_1, \cdots, v_m)$ captures the node labels or degrees, and build the Schur neuron $\mathfrak{n}_S$ from $L'$.*

## C  Related works

Higher order MPNNs have become a popular research area in graph representation learning since the seminal work by [32], which characterized the space of equivariant map w.r.t. $\mathbb{S}_n$ for $k$'th order tensors. The following works have mainly been divided into two streams, the one is based on $k$-Weisfeiler Leman test ($k$-WL test) [39, 19], for instances, [31] proposed $k$-order graph neural networks that is as powerful as $k$-WL test; [34] proposed $k$-GNN to approximate (or simulate) $k$-WL test; [14] further designed a $(k, t)$-FWL+ hierarchy that extends the $k$-WL test and implemented the corresponding neuron version. A common feature of these approaches is they all work on all $k$-tuples of vertices for $k$-th order neural networks and thus make their space and time complexity at least $\theta(n^k)$.

The other line of work is seeking ways to choose subgraphs in the graph, that are representative or contain important information, such as functional groups in a molecule. [7] chooses simplicial complex and [6] further extends this to any cell complex such as cycles. [21] incorporates any subgraph template and uses the equivariant maps defined [32] for local feature transform and devised a high order message passing scheme between those subgraphs, called *P*-tensor. This is arguably the most general framework in this line of work. This line of work is kind of independent of the $k$-WL test since the $k$-WL test is aimed to distinguish any pair of non-isomorphic graphs while chosen subgraphs are meant to work in some specific regions with domain prior. It's possible to still compare this kind of network with the distinguishing power of $k$-WL test, for example, [6] shows if they include cycles of size at most $k$ (including nodes and edges), their network are as powerful as WL test. However, such comparisons are not very meaningful to indicate the expressive power of such a domain-specific approach. In our opinion, the ability to learn certain features (such as count cycles of certain sizes or learn some function related to specific functional groups) might be a better criterion. It'll be an interesting research direction to design suitable criteria for the expressive power of such higher order MPNNs in general.

Our work follows this line of work since we're choosing specific subgraphs to learn representation on. But we utilize the subgraph's automorphism group to devise more possible equivariant maps (none of the aforementioned methods are built on the automorphism group). The only other work to our knowledge that uses the automorphism group is Autobahn [37], which is the first to introduce equivariance to the automorphism group to GNNs. The difference between our work and it lines twofold: (1) Autobahn theoretically introduces equivariant maps w.r.t. automorphism group via generalized convolution and Cosets, which is another group theoretical approach to construct equivariant maps, while we're using the decomposition to *irreps* (2) More importantly, Autobahn didn't mention a practical approach to construct those equivariant maps explicitly, hindering the utilization of more diverse subgraphs. In their experiments, they only use cycles of length five and six and paths of length three to six and construct the equivariant maps potentially by hand. In contrast, we give a simple and efficient way to construct equivariant maps w.r.t. automorphism group by EVD, which is very general to be used by any subgraph. Though our approach doesn't give the full set of equivariant maps, experiments show improvement over the traditional approach where only equivariant maps w.r.t. $S_n$ are used. We believe this algorithm, together with the group theoretical idea about decomposition $R^{m^k}$ into stable subspaces and *irreps* are beneficial to the research community.

There is a "third" type of higher order GNNs that is called Subgraph GNN, which deviates from the original definition of higher order MPNNs but can still be considered as higher order network. In particular, node-based GNNs such as [44, 46, 4, 42, 17] associate each node with a subgraph (by deleting the node, marking the node or extracting the EGO-network) and do MPNN on each subgraph,

where they get the resulting representation $X \in R^{n \times n}$ that can be considered as a 2nd order tensor representation on the original graph.

## D    Analysis of *Schur* layer

First of all, we notice that the equivariant map characterized in 1 (or 3) may not be the full set of equivariant maps w.r.t. $Aut_S$. Because (1) In the decomposition of $U = R^{m^k}$ to eigenspaces $U = U_1 \oplus \ldots U_t$, while $U_i$ is stable subspaces, it might not be *irreducible* and finer decomposition may be possible. In other words, there might be two irreps (isomorphic or not) corresponding to the same eigenvalue. (2) The maps defined by 1 didn't take into account the isomorphic subspaces corresponding to the same type of *irrep*, thus ignored the possible equivariant maps between $V_j^i$ and $V_{j'}^i$. So, it is of interest to find out how much the gap would be between our approach and the group theoretical approach. We first look at some examples in the first-order case (see table 4).

| Graph | $Aut_S$ | # of distinct Eigenvalues (*Schur* Layer) | $\frac{\sum_i (\kappa_i)^2}{\textit{irreps} \text{ approach}}$ | $\sum_i \kappa_i$ |
|---|---|---|---|---|
| 6-cycle | $D_6$ | 4 | 4 | 4 |
| 5-cycle | $D_5$ | 3 | 3 | 3 |
| 4-cycle | $D_4$ | 3 | 3 | 3 |
| 3-cycle | $D_3$ | 2 | 2 | 2 |
| 5-star | $S_4$ | 3 | 5 | 3 |
| 4-star | $S_3$ | 3 | 5 | 3 |
| 3-path | $S_2$ | 3 | 5 | 3 |
| n-cliques | $S_n$ | 2 | 2 | 2 |
| 5-cycle with one branch | $S_2$ | 6 | 20 | 6 |
| 6-cycle with one branch | $S_2$ | 7 | 29 | 7 |

Table 4: Examples on EVD approach vs group theoretical approach towards # of equivariant maps w.r.t. $Aut_S$. To calculate the decomposition of $R^m$ into *irreps*, one can calculate $\kappa_i = (\phi|\chi_i)$ where $\phi$ and $\chi_i$ is the character of the first-order action and ith *irrep* respectively, and $(|)$ is inner product. Another quick approach is to use $\phi(\sigma) = \sum_k \kappa_k \chi_k(\sigma)$ and look at some examples of $\sigma$ to determine what the $\kappa_k$ should be.

We note that for cycles in the graph case, there's no gap. However, when we add branches to the cycle to make the automorphism group smaller, the multiplicities of the *irreps* increase, e.g., in 5-cycle with one branch case, $S_2$ only have two 1-dimensional *irreps*: trivial and sign representation of permutation, and the first-order action decompose to 4 copies of trivial and 2 copies of sign representation, gives in total $4 * 4 + 2 * 2 = 20$ possible equivariant maps. However, note that # of *irreps* (counting multiplicities given by $\sum_i \kappa_i$) is equal to # of distinct eigenvalues, meaning that each eigenspace corresponds to an irreducible subspace and the decomposition of $R^m$ provided by eigenspaces is indeed a decomposition to the *irreps* in all of the cases listed in the table. Therefore, the multiplicities are the key reason for the gap between our EVD approach and the group theoretical approach, since our EVD approach can't capture the isomorphic property between subspaces. In principle, it is possible to find out which eigenspace is isomorphic to which, but in our current implementation, we didn't take this into account because we found out that only considering cycles is enough for a very good performance in the datasets we used.

Generally, it's complicated to determine the gap between our EVD approach and the group theoretic approach, especially in higher-order cases (where merely determining the $\kappa_i$'s is tricky), so we leave this into feature exploration.

## E    A brief overview about *P*-tensor framework

The *P*-tensor framework is a framework for linear equivariant maps w.r.t. $S_n$ both between the same subgraph and across different subgraphs. It is built on the equivariant maps characterized by [32].

**Definition 1** (*P-tensors*). *Let $U$ be a finite set of atoms and $D = (x_1, \ldots, x_d)$ an ordered subset of $U$. We say that a $k$-th order tensor $T \in \mathbb{R}^{d \times d \times \cdots \times d}$ is a $k$-th order permutation covariant tensor (or P-tensor for short) with reference domain $D$ if under reordering by $\tau \in S_d$ $D$ it transforms to*

$$[\tau \circ T]_{i_1, i_2, \ldots, i_k} = T_{\tau^{-1}(i_1), \ldots, \tau^{-1}(i_k)}.$$

### E.1 Message passing between *P*-tensors with the same reference domain

Consider the equivariant maps sending $T$ on $D$ to $T^{\text{out}}$ on $D$. The space of equivariant maps w.r.t $S_m$ is characterized by [32]:

**Proposition 1.** *The space of linear maps $\phi : \mathbb{R}^{dk_1} \to \mathbb{R}^{dk_2}$ that is equivariant to permutations $\tau \in S_d$ is spanned by a basis indexed by the partitions of the set $\{1, 2, \ldots, k_1 + k_2\}$.*

Then the authors designed a straightforward way to write the maps explicitly. Specifically, for each partition of the set $\{1, 2, \ldots, k_1 + k_2\}$, there are three parts that determines the equivariant map: (1) summing over specific dimensions or diagonals of $T^{\text{in}}$ (2) transferring $T^{\text{in}}$ to $T^{\text{out}}$ by identifying indices of $T^{\text{in}}$ with indices of $T^{\text{out}}$ (3) broadcasting the result along certain dimensions of $T^{\text{out}}$. These three operations correspond to the three different types of sets that can occur in a given partition $\mathcal{P}$: (1) those that only involve the second $k2$ numbers, (2) those that involve a mixture of the first $k_1$ and $k_2$ and (3) those only involve the first $k_1$ numbers. The type (1) - (3) corresponds to type (1) - (3) of operations with the dimensions in the sets.

For example, in the case $k_1 = k_2 = 3$, the $\mathcal{P} = \{\{1, 3\}, \{2, 5, 6\}, \{4\}\}$ partition corresponds to (a) summing $T^{\text{in}}$ along its first dimension (corresponding to $\{4\}$) (b) transferring the diagonal along the second and third dimension of $T^{\text{in}}$ to the second dimension of $T^{\text{out}}$ (corresponding to $\{2, 5, 6\}$) (c) broadcasting the result along the diagonal of the first and third dimensions (corresponding to $\{1, 3\}$). Explicitly, this gives the equivariant map:

$$T^{\text{out}}_{a,b,a} = \sum_c T^{\text{in}}_{c,b,b}$$

See table 5 for another example for $k_1 = k_2 = 2$ case.

| $\mathcal{P}$ | $\phi$ | $\mathcal{P}$ | $\phi$ |
|---|---|---|---|
| $\{\{1\}, \{2\}, \{3\}, \{4\}\}$ | $T^{\text{out}}_{a,b} = \sum_{c,d} T^{\text{in}}_{c,d}$ | $\{\{2\}, \{1,3,4\}\}$ | $T^{\text{out}}_{b,a} = T^{\text{in}}_{b,b}$ |
| $\{\{1\}, \{2\}, \{3,4\}\}$ | $T^{\text{out}}_{a,b} = \sum_c T^{\text{in}}_{c,c}$ | $\{\{1,2,3\}, \{4\}\}$ | $T^{\text{out}}_{a,a} = \sum_b T^{\text{in}}_{a,b}$ |
| $\{\{1\}, \{2,4\}, \{3\}\}$ | $T^{\text{out}}_{a,b} = \sum_c T^{\text{in}}_{c,b}$ | $\{\{1,2,4\}, \{3\}\}$ | $T^{\text{out}}_{a,a} = \sum_b T^{\text{in}}_{b,a}$ |
| $\{\{1\}, \{2,3\}, \{4\}\}$ | $T^{\text{out}}_{a,b} = \sum_c T^{\text{in}}_{b,c}$ | $\{\{1,2\}, \{3,4\}\}$ | $T^{\text{out}}_{a,a} = \sum_c T^{\text{in}}_{c,c}$ |
| $\{\{2\}, \{1,4,3\}\}$ | $T^{\text{out}}_{b,a} = T^{\text{in}}_{b,b}$ | $\{\{1,3\}, \{2,4\}\}$ | $T^{\text{out}}_{a,b} = T^{\text{in}}_{a,b}$ |
| $\{\{1,3\}, \{2,4\}\}$ | $T^{\text{out}}_{b,a} = \sum_c T^{\text{in}}_{c,b}$ | $\{\{1,4\}, \{2,3\}\}$ | $T^{\text{out}}_{b,a} = T^{\text{in}}_{b,a}$ |
| $\{\{1,2,3,4\}\}$ | $T^{\text{out}}_{a,a} = T^{\text{in}}_{a,a}$ | $\{\{1,2,3,4\}\}$ | $T^{\text{out}}_{a,a} = T^{\text{in}}_{a,a}$ |

Table 5: The $\mathcal{B}(4) = 15$ possible partitions of the set $\{1, 2, 3, 4\}$ and the corresponding permutation equivariant linear maps $\phi : \mathbb{R}^{k \times k} \to \mathbb{R}^{k \times k}$.

### E.2 Message passing between *P*-tensors with the different reference domains

If $T^{\text{in}}$ has reference domain $D_1$ and $T^{\text{out}}$ has $D_2$, with $D_1 \neq D_2$ and $D_1 \cap D_2 \neq \emptyset$. We could have more options corresponding to summing either over the intersection or over $D_1$, and broadcasting either over the intersection or over $D_2$. So the number of maps corresponging to partition of type $(p_1, p_2, p_3)$ is $2^{(p_1 + p_3)}$. Let $D_1 \cap D_2 = d^\cap$, the previous example would have the following maps in $D_1 \neq D_2$ case:

$$T^{\text{out}}_{a,b,a} = \begin{cases} \sum_{c=1}^{d^\cap} T^{\text{in}}_{c,b,b} & a, b \leq d^\cap \\ 0 & \text{otherwise,} \end{cases} \quad (a \in \{1, \ldots, d^\cap\}, \ c \in \{1, \ldots, d^\cap\})$$

$$T^{\text{out}}_{a,b,a} = \begin{cases} \sum_{c=1}^{d^\cap} T^{\text{in}}_{c,b,b} & b \leq d^\cap \\ 0 & \text{otherwise,} \end{cases} \quad (a \in \{1, \ldots, d_2\}, \ c \in \{1, \ldots, d^\cap\})$$

$$T^{\text{out}}_{a,b,a} = \begin{cases} \sum_{c=1}^{d_1} T^{\text{in}}_{c,b,b} & a, b \le d^{\cap} \\ 0 & \text{otherwise,} \end{cases} \quad (a \in \{1, \dots, d^{\cap}\}, \ c \in \{1, \dots, d_1\})$$

$$T^{\text{out}}_{a,b,a} = \begin{cases} \sum_{c=1}^{d_1} T^{\text{in}}_{c,b,b} & b \le d^{\cap} \\ 0 & \text{otherwise,} \end{cases} \quad (a \in \{1, \dots, d_2\}, \ c \in \{1, \dots, d_1\})$$

# F Group Representation Theory Background

Here, we explain in detail the decomposition of $k$-th order permutation to *irreps* of $S_m$.

**Proposition 2.** *Let* $\rho_\lambda, \lambda \vdash m$ *be the* irreps *of* $S_m$, $U = R^{m^k}$, $\sigma \in S_m$ *act on* $U$ *in the manner of equation 7, and suppose this action contains* irreps $\rho_{\lambda_1}, \dots, \rho_{\lambda_p}$ *with multiplicities* $\kappa_1, \dots, \kappa_p$, *we have:*

$$U = U_1 \oplus U_2 \oplus \dots \oplus U_p, \quad U_i = \bigoplus_j^{\kappa_i} V_i^j$$

*where the first part gives the canonical decomposition of* $U$ *and the second part further decompose each* $U_i$ *into irreducible subspaces, where* $V_i^j$ *correspond to* $\rho_{\lambda_i}$. *In matrix form,*

$$P^{(k)}_\sigma = M \begin{pmatrix} \rho_{\lambda_1}(\sigma) & 0 & \cdots & 0 \\ 0 & \rho_{\lambda_2}(\sigma) & \cdots & 0 \\ \vdots & \vdots & \ddots & \vdots \\ 0 & 0 & \cdots & \rho_{\lambda_p}(\sigma) \end{pmatrix} M^T, \quad \text{for all } \sigma \in S_m$$

*where* $M$ *is the orthogonal transformation for basis and we abuse* $\rho_{\lambda_i}(\sigma)$ *to denote also the matrix of* $i$-*th* irrep *in the decomposition.*

If we take a closer look at $M$, we can find that since $P^{(k)}_\sigma$ is under the standard basis of $R^{m^k}$, thus $M$ is just the orthonormal basis of each $U_i$ (thus each $V_i^j$) combined together, we denote $M = (M_1, \dots, M_p)$ with $M_i$ a $m^k \times (d_{\lambda_i} * \kappa_i)$ dimensional matrix, where $d_{\lambda_i}$ is the degree of $\rho_{\lambda_i}$. Therefore, $P^{(k)}_\sigma T$ becomes:

$$P^{(k)}_\sigma T = [M_1, M_2, \dots, M_p] \begin{pmatrix} \rho_{\lambda_1}(\sigma) & 0 & \cdots & 0 \\ 0 & \rho_{\lambda_2}(\sigma) & \cdots & 0 \\ \vdots & \vdots & \ddots & \vdots \\ 0 & 0 & \cdots & \rho_{\lambda_p}(\sigma) \end{pmatrix} \underbrace{\begin{pmatrix} M_1^T \\ M_2^T \\ \vdots \\ M_p^T \end{pmatrix}}_{(1)} T \tag{10}$$

$$\underbrace{\phantom{\hspace{10cm}}}_{2}$$

The part (1) is a generalized Fourier transform of $T$ to its Fourier components (coordinates under the orthonormal basis) $\hat{T} = \begin{pmatrix} M_1^T T \\ M_2^T T \\ \vdots \\ M_p^T T \end{pmatrix} \triangleq \begin{pmatrix} B_1 \\ B_2 \\ \vdots \\ B_p \end{pmatrix}$, and the part (2) is the *irreps* $\rho_{\lambda_1}, \dots, \rho_{\lambda_p}$ act

independently on $\hat{T}$ with each component $B_i \mapsto \begin{pmatrix} \rho_{\lambda_i}(\sigma) & \cdots & 0 \\ \vdots & \ddots & \vdots \\ 0 & \cdots & \rho_{\lambda_i}(\sigma) \end{pmatrix} B_i$. Note that there're $\kappa_i$

multiple of $\rho_{\lambda_i}$ act the same on components of $B_i$ correspond to $V_i^j$, with a slight abuse of notation, we can think $B_i \in R^{d_{\lambda_i} \times \kappa_i}$ (instead of $R^{(d_{\lambda_i} * \kappa_i) \times 1}$) and write this map as $\rho_{\lambda_i}(\sigma) B_i$, the fact that the map by *irrep* $\rho_{\lambda_i}$ act independently on each column of $B_i$ allow us to identify directly a set of equivariant maps by multiplying $B_i$ with matrix $W_i \in R^{\kappa_i \times \kappa_i}$ to the right, and using associativity of matrix multiplication: $\rho_{\lambda_i}(\sigma)(B_i W_i) = (\rho_{\lambda_i}(\sigma) B_i) W_i$. This gives in total of $\sum_i \kappa_i^2$ independent equivariant maps, as stated in the main text. The last part of the above equation maps Fourier components to their original space. In short, we can write:

$$P^{(k)}_\sigma T = \sum_i M_i \rho_{\lambda_i}(\sigma) \underbrace{M_i^T T}_{B_i} \tag{11}$$

with the abuse of notation to rearrange elements in $B_i$ mentioned above.

Then we present a detailed version of theorem 2 in the main text.

**Theorem 4** (Necessary and sufficient condition for equivariant map)*. Let $G$ be a finite group acting on a vector space $U$ by the linear action $\{g : U \to U\}_{g \in G}$ and assume the action can be decomposed into* irreps $\rho_1, \ldots, \rho_p$ *with multiplicities* $\kappa_1, \ldots, \kappa_p$ *and degree* $d_i$:

$$U = U_1 \oplus U_2 \oplus \cdots \oplus U_p, \quad U_i = \bigoplus_{j=1}^{\kappa_i} V_i^j$$

*Then $\phi : U \to U$ is an equivariant map w.r.t. this group action if and only if $\phi$ is of the form:*

$$\phi(v) = \sum_{j'} \alpha_{j,j'}^i \tau_{j \to j'}^i(v) \quad \text{for } v \in V_j^i \tag{12}$$

*for some fixed set of coefficient $\{\alpha_{j,j'}^i : i \in [1, \ldots, p], j, j' \in [1, \ldots, \kappa_i]\}$.*
*In matrix form, suppose the matrix of $g$ is $R_g$ under basis $(e_i)$ and $\dim(U) = n$, and it decompose into:*

$$R_g = M \begin{pmatrix} \rho_1(g) & 0 & \cdots & 0 \\ 0 & \rho_2(g) & \cdots & 0 \\ \vdots & \vdots & \ddots & \vdots \\ 0 & 0 & \cdots & \rho_p(g) \end{pmatrix} M^T \tag{13}$$

$$= \sum_i M_i \rho_i(g) M_i^T \tag{14}$$

*Then $\phi : R^n \to R^n$ is equivariant if and only if it is of the form:*

$$\phi(T) = M M^T T \begin{pmatrix} W_1 & 0 & \cdots & 0 \\ 0 & W_2 & \cdots & 0 \\ \vdots & \vdots & \ddots & \vdots \\ 0 & 0 & \cdots & W_p \end{pmatrix} \tag{15}$$

$$= \sum_i M_i \underbrace{M_i^T T}_{B_i} W_i \tag{16}$$

*where $T \in R^n$, $W_i \in R^{\kappa_i \times \kappa_i}$ is the fixed coefficients and $B_i$ rearrange to $R^{d_i \times \kappa_i}$ when needed. The coefficients $\alpha_{j,j'}^i$ corresponds to $(W_i)_{j,j'}$.* [3]

*Proof.* **Sufficiency:** firstly, $\alpha_{j,j'}^i \tau_{j \to j'}^i(v)$ is equivariant by definition of isomorphism map. And the equivariance of $\phi$ follows from the fact that sum of equivariant maps are equivariant.

**Necessity:** consider $\phi$ on $V_j^i$ and let $W = Im(\phi|_{V_j^i})$ the image of $\phi$ on $V_j^i$. Since $\phi \circ g = g \circ \phi$ and $g(v) \in V_j^i$ for any $v \in V_j^i$, we have $g(\phi(v)) = \phi(g(v)) \in W$, so $W$ is stable under the action. Thus we can decompose $W$ into irreducible spaces $W = W_1 \oplus \ldots W_p$. Apply this decomposition to $\phi(v)$ for $v \in V_j^i$, we get:

$$\phi(v) = \phi_1(v) + \ldots + \phi_p(v) \quad \text{where } \phi_i(v) \in W_i$$

In other words $\phi_k = Proj_k \circ \phi$ where $_k$ is the projection of $W$ onto $W_k$ (uniquely defined by the direct sum decomposition). Then $\phi_k : V_j^i \to W_k$ and $\phi_k \circ g = g \circ \phi_k$. By Schur's lemma [36], for $\phi_k \neq 0$, we must have $W_k$ isomorphic to $V_j^i$ and $\phi_k(v) = \theta_k \tau_{V_j^i, W_k}(v)$. Since only $V_{j'}^i$ is isomorphic to $V_j^i$, we have $W_k = V_k^i$ and $\phi_k(v) = \alpha_{j,k}^i \tau_{j,k}(v)$, where $\tau_{j,k}(v)$ is the isomorphic map sending $V_j^i$ to $V_k^i$. Thus $\phi(v) = \sum_k \phi_k(v) = \sum_k \alpha_{j,k}^i \tau_{j,k}(v)$ for $v \in V_j^i$.

Finally, by linearity and $U = \bigoplus_{i,j} V_j^i$, the only possible equivariant function $\phi : U \to U$ is given by 12.

---

[3]Similar discussions could be found in Section 4 of [38], but without a proof for necessity.

**Connection to matrix form:** first note that the isomorphism $\tau^i_{j\to j'} : V^i_j \to V^i_{j'}$ is given by:

$$\tau^i_{j\to j'}(u^i_{j,l}) = u^i_{j',l}$$

where $\{u^i_{j,l}, l = 1, \ldots, \kappa_i\}$ and $\{u^i_{j',l}, l = 1, \ldots, \kappa_i\}$ are orthonormal basis for $V^i_j$ and $V^i_{j'}$ respectively such that the action of $g$ is associated with matrix $\rho_i(g)$. Therefore,

$$\phi(u^i_{j,l}) = \sum_{j'} \alpha^i_{j,j'} \tau^i_{j\to j'}(u^i_{j,l})$$

$$= \sum_{j'} \alpha^i_{j,j'}(u^i_{j',l})$$

Thus the matrix of $\phi$ under basis $(u^i_{j,l}, i = 1, \ldots, p, j = 1, \ldots, \kappa_i, l = 1, \ldots, d_i)$ is

$$\begin{pmatrix} W_1 & & & & & \\ & \ddots & & & & \\ & & W_2 & & & \\ & & & \ddots & & \\ & & & & W_p & \\ & & & & & \ddots \end{pmatrix}$$

(which is the same as 
$$\begin{pmatrix} W_1 & 0 & \cdots & 0 \\ 0 & W_2 & \cdots & 0 \\ \vdots & \vdots & \ddots & \vdots \\ 0 & 0 & \cdots & W_p \end{pmatrix}$$
with $B_i$ rearrange to $R^{d_i \times \kappa_i}$) and $M$ is just the matrix to transform standard basis $(e_i)$ to basis $(u^i_{j,l})$.      $\square$

# G    Ways to Use Schur Neuron

While Schur neuron can be viewed as a generic transform on any subgraph's first-order activation, we provide some thoughts and possibilities to instantiate it.

In our experiment, we primarily view Schur neuron as a way to expand the feature of a subgraph. It can be viewed as an analogue to CNN's convolution filter and it's indeed a constrained version of spectral convolution filter [5] on the subgraph. In light of this, we call number of channels of Schur neuron as how many times it expand the output feature versus the input feature.

Besides, we observe that the linear equivariant map described [32] is a special case of Schur neuron's map. The two possible linear map in [32] is (1) identity map, corresponding to take all weights $W_i = 1$ (2) $T \to \Sigma_i T_i \mathbb{1}$, where $\mathbb{1}$ is all ones vector, corresponding to set $W_1 = 1$, and $W_i = 0$ for $i \neq 1$ since $\mathbb{1}$ is eigenvector of any graph Laplacian with eigenvalue $0$. Consequently, we suggest to keep the two basic but important linear maps and augment them with Schur neuron. Empirically we verified this by an experiment only vary the number of channels. In figure 1, we see that further increase the number of channels beyond 4 wouldn't give us performance gain since cycle 5 and 6 only has 3 and 4 distinct eigenvalues respectively.

Furthermore, we suggest that number of channels of Schur neuron be proportional or approximately equal to number of distinct eigenvalues of the subgraph it attach to. This is because the later is the number of independent linear maps for Schur neuron.

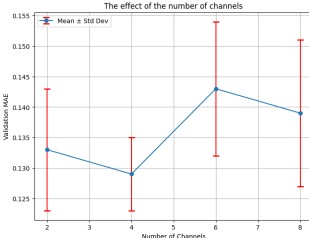

Figure 1: A study on num of channels in Schur layer. Cycle 5 and 6 are included.

Additonally, we're wondering if we can use *Schur* layer in place of MLP's to provide a local structure aware transform to subgraph activations. Therefore, we tried to replace the 2-layer MLP with 2-layer

*Schur* layer in the original *Linmaps* architecture. But it turns out this use case wasn't as effective as the use as learning new feature. Possibly because *Schur* layer learns feature itself while MLP transform the learned feature to some desired space.

| Methods | Validation MAE |
|---|---|
| *Linmaps* | 0.087 |
| *Schur* layer (in place of MLP) | 0.081 |
| *Schur* layer (as learning new feature) | 0.076 |

Table 6: Comparison about two use cases of *Schur* layer. Experiment on ZINC-12K.

Finally, regarding flexibility as discussed 5.2, in table 7, we see that adding cycles with branches could provide the *Schur* Net with more detailed topological information, thus improving the performance. And the code for adding this actually only requires a single definition of those templates without modification to the neural network.

| Model | Validation MAE |
|---|---|
| *Schur*-Net on 5,6 cycles(baseline) | $0.113 \pm 0.008$ |
| *Schur*-Net on 5,6 cycles with up to three branches | $\mathbf{0.105 \pm 0.001}$ |

Table 7: Flexibility of *Schur* layer. Experiments on ZINC-12k dataset. All other settings are the same. A smaller network than previous experiments was used.

# H   More experiment results

We first start with some molecular datasets in TUdataset [33], which is a small but commonly used benchmark for GNN. In table 8, we see that *Schur* layer consistently improves over *Linmaps* in various bioinformatics and molecule datasets.

| **Dataset** | *Linmaps* | *Schur* Layer |
|---|---|---|
| Proteins | $74.7 \pm 3.8$ | $\mathbf{75.4 \pm 4.8}$ |
| MUTAG | $89.9 \pm 5.5$ | $\mathbf{90.94 \pm 4.7}$ |
| PTC_MR | $61.1 \pm 6.9$ | $\mathbf{64.6 \pm 5.9}$ |
| NCI1 | $82.1 \pm 1.8$ | $\mathbf{82.7 \pm 1.9}$ |

Table 8: Comparison of *Linmaps* and *Schur* Layer performance on TUdatasets. Numbers are binary classification accuracy.

In table 9, we compare the runtime of our *Schur* layer and *Linmaps*, which shows the extra computational cost of *Schur* layer wasn't significant while being able to use more equivariant maps and achieving better accuracy.

| **Dataset** | *Linmaps* | *Schur* Layer |
|---|---|---|
| Zinc-12k | 25.4s | 27.6s |
| NCI1 | 9.5s | 11.5s |

Table 9: Runtime per epoch with hyper-params num_layers = 4, rep_dim = 128, dropout = 0.0, batch_size = 256, num of channels = 4, cycle_sizes = 3,4,5,6,7,8. This shows *Schur* Layer didn't add much computational costs to Linmaps while being more expressive.

Another ablation study is perform on ZINC-12K dataset. In table 10, we see that *Schur* layer outperforms *Linmaps* when certain cycle sizes are considered, especially when only cycles 5 and 6 are chosen as subgraphs in the neural network.

| Layer | cycle size {5,6} | cycle size {3,4,5,6} |
|---|---|---|
| *Linmaps* (baseline) | $0.139 \pm 0.007$ | $0.103 \pm 0.011$ |
| *Schur* layer | $\mathbf{0.114 \pm 0.014}$ | $\mathbf{0.100 \pm 0.009}$ |

Table 10: Comparison between *Schur* Layer and *Linmaps* with different set of cycles chosen. Experiments on ZINC-12k dataset and all scores are test MAE.

# I  Architecture and experiment details

In designing of our architecture, we design a message passing scheme similar to [21] and add Schur layer to each of the convolution layers. A visualization of our architecture can be found in Figure 2. In each convolution layer, we maintain 0th-order node representation ($h_v \in R^{|V|*dim}$) and edge representation ($h_e \in R^{|E|*dim}$) where we assume the input graph $G = (V, E)$. We further maintain 1st-order representation on cycles ($h_{D_k} \in R^{(k*(c_k)*dim)}$) where $D_k$ is cycle of length k and $c_k$ is the number of such cycle in $G$. We did message passing between node and edge representation the same as CIN [6]. The edge and cycle pass message with each other by tranfer maps described by [21] and then the incoming message to cycles as well as the original cycle representation is transformed by Schur layer. When updating the edge representation and cycle representation, we combine the original representation with the incomming message by either concatenation or $\epsilon$-add described in GIN [40] and feed it into an MLP to get new representations.

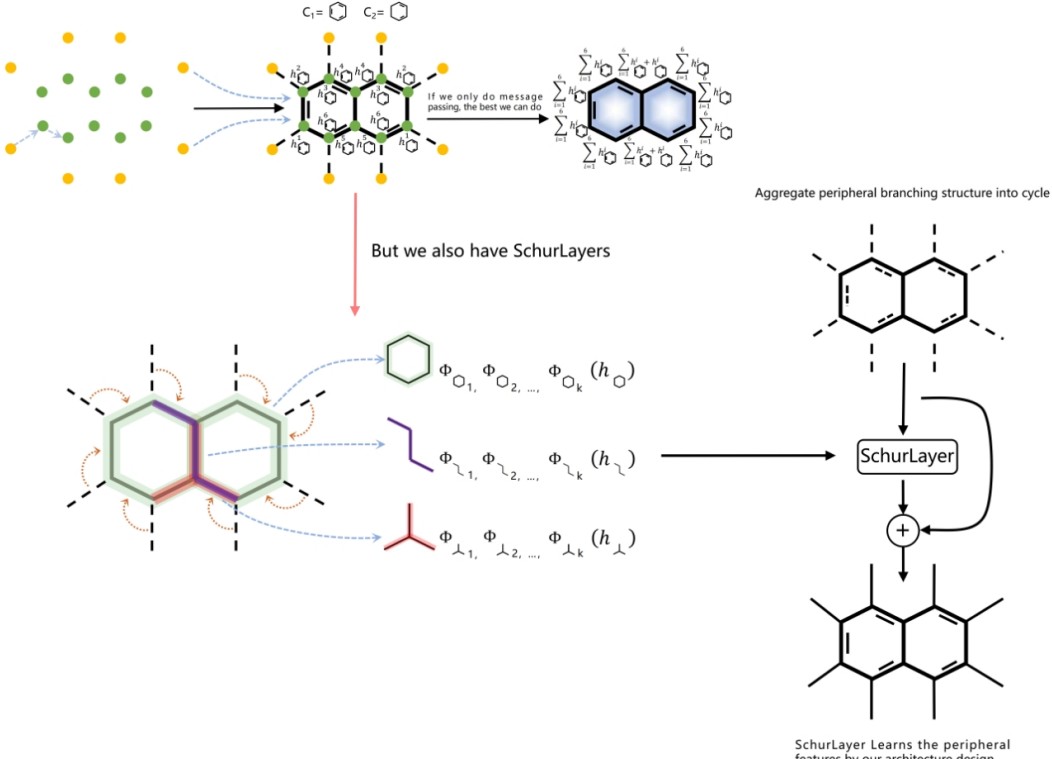

Figure 2: An example illustration of one layer of our model architecture. For an input graph, we first execute the standard node and edge level message passing and lift up the model to higher-order by forming higher-order representations on specific template graphs. In this example, we learned the representation on a Naphthalene. To further decouple its information, we may apply the SchurLayer on the template graphs that would potentially provide us insight about the graph. In this example, the 6 cycle, star graph, and path graph are all applying SchuLayer to learn the features locally. Through our architecture design, we can also incorporate the peripheral information. Automorphism group then helps us discover distinct key features such as non-symmetries, and we pass these aggregated features into the next layer of the architecture.

**Hyperparameters**    For experiments on ZINC-12K and OGB-HIV, we tune representation dimension in $\{32, 64, 128\}$, and experiment on cycles up to length 18. In the best-performing model, we use representation dimension 128 and cycle lengths from 3 to 8 and number-of-layers is always 4. For *Schur* layer, we tuned number of channels from 2 to 8 and found $2, 4$ are a suitable choice when it is used as an augmentation to *Linmaps*. For MLPs, we either use 2 to 3 layers depending on how dense the input's information is. We always use a batchnorm [25] after the Linear layer and then do ReLu activation. For training hyperparameters, we use an init learning rate of 0.01 and use ReduceLROnPlateau scheduler in PyTorch with patience of 10 or 30. We use Adam optimizer [27] for all trainings. We train for 500 or 1000 epochs depending on model size. In particular, in table 10, the models have representation dimension 32, so it is trained to only 500 epochs. All other models are trained 1000 epochs. A batch size of 256 is used for all models. All results are run at least three times to calculate the standard deviation.

For experiments on TUdataset (table 8), we chose a set of hyper-params by intuition (we didn't tune them because this experiment is to demonstrate the effectiveness of *Schur* layer and compare *Schur* layer with *Linmaps* under the same experiment setting. we don't aim to compare with Sota on this experiment). Hyper-params for both *Schur* Layer and *Linmaps*: num_layers = 4, rep_dim = 32,64, dropout = 0.5, batch_size = 32,128, lr = 0.01, num of channels = {2,4}, cycle sizes = 3,4,5,6. We're training with StepLR scheduler where reduce lr to 1/2 after every 50 epochs, with a total of 300 epochs. The hyperparams weren't tuned, we just chose a smaller value for a smaller dataset and a bigger value for bigger datasets by rule-of-thumb. Linmaps is implemented on our own according to the description of *P*-tensor, then *Schur* Layer replaces Linmaps operation by *Schur* operation. We follow the evaluation strategy specified in [40].

**Used Compute Resources**    Experiments are all run on one Tesla L4 from PyTorch Lightning Workspace and one NVIDIA RTX 4090. The code is implemented in PyTorch. For the running time, on the ZINC-12k dataset, it takes around $8.53 \pm 1.2$ hours to finish training for 1000 epochs with 4 layers and 128 dimension representation on an NVIDIA RTX 4090. We're running on a desktop with a Ryzen 7900 CPU and 64GB of RAM.

