# OpenReview forum: "Schur Nets: exploiting local structure for equivariance in higher order graph neural networks"
_NeurIPS.cc/2024/Conference — NeurIPS 2024 poster_

### Official Review · Reviewer_uvzr · 2024-06-25

**Soundness:** 3
**Presentation:** 2
**Contribution:** 3
**Rating:** 5
**Confidence:** 2

**Summary:**

This paper proposes Schur layers. Schur layers are meant to be used in higher-order MPNNs and are based on respecting local automorphism equivariance, however without the need for explicitly computing all automorphisms. In experiments, this method achieves state-of-the-art results on ZINC.

**Strengths:**

- **(S1 - Significance)** Schur Layer are a general  architecture and can thus be used in a lot of higher-order architecture. This means that (theoretically) interesting properties of Schur Layers can be used to improve many other architectures

- **(S2 - Significance)** In experiments on the ZINC dataset, SchurNets seem to consistently outperform other methods.

- **(S3 - Novelty)** The approach does seem to be entirely novel, albeit based on some well-known mathematics.

- **(S4 - Clarity & Quality)** Sections 1 and 2 are very well written. Especially Section 2 which introduces MPNNs based on the way the authors understand equivariance presents an extremely interesting way of thinking about GNNs.

**Weaknesses:**

- **(1 - Clarity)** The theorems are difficult to understand and no intuitive explanation is given. To me it is not clear what the theoretical contributions of this paper mean.
- **(W2 - Significance)** It seems to me, as if the theoretical contributions are quite small. However, I am unable to really verify this due to (W1).
- **(W3)** The paper is not reproducible as the code is not supplied (“ We will also make our code publicly accessible if the paper get accepted”).
- **(W4 - Quality)** The experiments are lacking:
	- **(W4.1)** All experiments are performed on a single dataset (ZINC12k). This dataset in particular has been used in a lot of GNN papers and by now the community is probably overfitting to this specific dataset.
	- **(W4.2)** The advantages of SchurNet are small often only making a difference in the third digit of the MAE (for example in Table 3 the difference between the best SchurNet achieves 0.064 \pm 0.004 and a higher order MPNN $0.071 \pm 0.02$, that is a difference of 0.007).
	- **(W 4.3)** Replacing layers in higher-order MPNNs with Schur-Layers is in my opinion the most exciting application of this approach. Unfortunately, the authors only experiment with this for a single model ($P$-tensors).


Overall, I think that SchurNets could be an interesting and useful method to be used with higher-order GNNs. However, I think  this paper requires more work and thus vote to reject. In particular: (1) the writing needs to be improved and better explanations are needed for the theory; and (2) the paper needs more experiments on diverse datasets with more models.

**Questions:**

- **(Q1)** What is the (intuitive) meaning behind: Theorem 1, Theorem 2 and Theorem 3?
- **(Q2)** As far as I understand it Schur Layers are equivariant under the automorphism. Does that mean that there is some limit to their expressivity? Otherwise, you would be solving the NP-hard automorphism problem?


## Miscellaneous
- **(M1)** In my opinion, Table 5 is important enough to your narrative to warrant being in the main paper.
- **(M2)** Missing citation in line 186.

**Limitations:**

Yes

---

> ### Author Rebuttal · Authors · 2024-08-06
>
> We thank the reviewer for their insightful comments on the paper.
>
> QUESTION 1. What is the intuition behind Lemma 1- theorem 3? The theory of equivariant neural nets
> usually proceeds by considering the action of a group on the space that the output of a given layer of the
> network lives in,  and decomposing this space into an orthogonal sum of subspaces that transform
> independently, i.e. subspaces that are invariant under the action. Then they find operators on each
> subspace that are invariant and show that any linear combination of these operators is a valid
> equivariant map. All this usually involves irredudible representations of the group, generalized
> Fourier transforms and other abstract mathematical tools.
>
> In contrast, Section 4.1 of our paper shows that in the case of equivariance to the automorphism group
> of subgraphs, essentially the same can be accomplished simply by finding the eigenspaces of the
> graph Laplacian. To the best of our knowledge, this has not been done before, and we believe that it is of independent interest.
> So, in summary, the intuition is that the outputs of the subgraph-layers decompose into subspaces that transform independently under elements of the automorphism group, the specific decomposition is reflective of structure of the automorphism group, but still, the decomposition (up to the the expressiveness gap discussed below) can be found simply from the graph Laplacian. We think this result is not obvious, but it can be visualized intuitively, and it is an important contribution of the paper. Following your comments, we will try to make Section 4.1 less dry and more intuitive. Specifically, some diagrams might help visualize the decomposition into subspaces.
>
> QUESTION 2. Yes and no. There is indeed an interesting connection to graph isomorphism but as we also
> mention in the response to Reviewer uFzV, it is not that simple. As we admit in the Limitations,
> Schur layers are not guaranteed to be the most general automorphism-equivariant linear layers.
> Specifically, following the proof of our theorems, there might be a gap in expressivity if the eigenvalues
> corresponding to distinct invariant subspaces coincide. However, the literature on graph isomorphism
> amongst others suggests that barring extra symmetries, for moderate sized graphs, this is quite exceptional.
> So for pratical purposes Schur layers are almost as good as the full group theoretical approach (which
> would be infeasible to implement in a practical GNN).
>
> Regarding using Schur layers for finding the automorphism group, that is unfortunately not so easy.
> Schur layers can generate automorphism equivariant maps without having to find the automorphism group
> explicitly. That does not mean that we could somehow reconstruct the automorphism group itself
> from the equivariant maps.
>
> FURTHER COMMENTS
>
> - We acknowledge that the experimental results are limited. We were pressed for time and limited in
> computational resources before the deadline. We have since conducted experiments on several other
> standard benchmarks and we see a consistent improvement when we add Schur layers (see global rebuttal).
>
> - The reason we mostly focus on comparing performance to P-tensors is that P-tensors [Hands et al, AISTATS 24] is currently the best performing (and in many ways most general) prior model amongst subgraph based higher order GNNs.
> Specifically on ZINC 12K, according to the results reported in [Hands et al, AISTATS 24] they represent the state of the art. Moreover, Schur layers work best in tandem with a broader higher order message passing architecture, and P-tensors are one of the most general such frameworks. So it is natural to focus our experiments on contrasting higher order message passing with or without Schur-layers.
>
> - Finally, it is not uncommon for the competition between GNNs to come down to just the 1% level in accuracy. This is a crowded field and experience shows that even very simple architectures can bring down the error to ~90% of the state of the art. Recent works focus on extracting more subtle information from graphs that is responsible for the last few percent of improvement. Admittedly, the competition on the standard benchmarks is getting saturated. In future work we want to apply Schur Nets to practical chemistry problems where the flexibility of the framework and its ability to capture chemically relevant structures (functional groups) will be the most important. First, however we need to validate the architecture by showing its performance on the benchmarks.
>
> - Reproducibility. As we explain the response to Reviewer uFzV, Schur layers themselves are not difficult
> to implement. However, because of the irregular nature of information flow between subgraphs, the rest of
> the higher order GNN framework is quite involved, especially if all the messages need to be computed
> in parallel on the GPU. This forced us to write a separate C++/CUDA software library for general higher
> order message passing with a PyTorch interface. The library is available on github, but we cannot share a
> link to it here without jeopardizing anonimity. Once the injunction for anonimity is lifted, the
> link to the library will be added to the paper and naturally we will also make the training scripts public.

---

> > ### Comment · Reviewer_uvzr · 2024-08-12
> >
> > I thank the authors for their answers and the thorough rebuttal. The additional experiments clearly strengthen this paper (especially the performance on MOLHIV). Furthermore, I find the explanations of the theorems really intriguing and am looking forward to seeing visualizations of them. Overall, if the authors manage to improve their explanations I am not against acceptance and have thus increased my score accordingly.

---

### Official Review · Reviewer_fYBT · 2024-07-01

**Soundness:** 2
**Presentation:** 3
**Contribution:** 3
**Rating:** 6
**Confidence:** 4

**Summary:**

This paper introduces Schur layers, a novel approach in graph neural networks (GNNs) that enhances expressive power by leveraging spectral graph theory. Traditional GNNs struggle with capturing complex local graph structures due to their reliance on full permutation equivariance, which is overly restrictive and computationally intensive. Schur layers address this issue by computing basis functions directly from the graph Laplacian, circumventing the need for explicit enumeration of automorphism groups. This method is shown to significantly improve GNN performance, particularly in scenarios like molecular data analysis where local subgraph structures (e.g., cycles) are crucial.

**Strengths:**

1. This paper makes a significant contribution by introducing Schur layers, which offer a novel approach to enhancing the expressive power of GNNs through higher order message passing.

2. The theoretical framework and mathematical rigor underpinning Schur layers are robust and well-explained.

**Weaknesses:**

1. Experimental Evaluation: Limited empirical evaluation on diverse datasets and benchmarks undermines broader applicability and comparison with existing methods.

2. Contextualization: The paper could better situate its contributions within the broader landscape of GNN research, particularly in comparison with other higher order message passing techniques.

**Questions:**

1. How does the computational complexity of Schur layers compare to traditional GNN approaches in terms of training time and memory usage?

2. How sensitive are the performance gains of Schur layers to different types of molecular datasets beyond the benchmarks used in the paper, and how would they perform in varied experiments using larger datasets such as OGB datasets and different tasks like link prediction to validate the model's versatility and resilience?

**Limitations:**

The authors have addressed several limitations, such as the spectral approach not ensuring the finest decomposition into invariant subspaces, potentially limiting the generality of automorphism-equivariant linear maps and the theoretical expressivity gap this might cause. However, a more detailed discussion of the method's limitations in terms of computational overhead and potential impact on large-scale graph datasets would be beneficial.

---

> ### Author Rebuttal · Authors · 2024-08-06
>
> QUESTION1 In short: it is not the Schur layers themselves that make this architecture more expensive than
> classical GNNs, but rather the higher order message passing itself (like in P-tensors).
> The natural thing to compare Schur-layers to are the vanilla "linmaps" operations.
> In the case of a subgraph with $m$ nodes represented with a 1st order permutation equivariant tensor with
> $c$ channels (i.e., $X \in R^{m*c}$) the cost of the Linmaps operation followed by a linear layer mixing
> the channels is $O(mc+mc^2)$, whereas the cost of the Schur Layer would be $O(m^2 c+ mc^2)$.
> Since the number of channels is typically much larger than the size of the subgraph,
> these are essentially the same.
>
> In general, higher order GNNs are more expensive because (a) there are potentially many more overlapping
> subgraphs than vertices (b) in the case of k'th order message passing between subgraphs of size m,
> the size of the messages scales with $m^k$ (c) the way that each pair of subgraphs communicates depends
> on exactly how many vertices and which vertices they overlap in, so the communication protocol cannot be
> reduced to a simple gather/scatter operation of the type that PyTorch Geometric performs.
> Since the subgraphs used in practice are still relatively small (think $m=6$) it is really the last point
> that constrains speed. To get around it we spent months architecting a C++/CUDA based higher order
> message passing library with which we could run our experiments only 3-4 times slower than conventional GNNs.
> Unfortunately we can't give a pointer to this library without jeopardizing anonymity.
>
> As for memory, this is again a general higher order GNN issue. Instead of just storing a $c$ element vector
> at each vertex, we now store $m\times c$ matrices of $m\times m\times c$ tensors. For the types of small
> molecules datasets that we mostly experimented with this is still not an issue. The practical issue  is that
> efficient higher order message passing on GPUs involves precomputing various control data structures and for
> large datasets we can't store all of these in the GPU's RAM, while moving information back and forth between
> the GPU and system memory is slow.
>
> QUESTION 2. We did some more experiments on TUdatasets and OGB-hiv datasets.
> Results are attached in general rebuttal. We observe consistent improvement by just adding Schur layer
> (replacing Linmaps) which shows the robustness of performance gain. Since Schur layers are strictly more expressive than layers which cannot take the local automorphism group into account (as a form of side-information), it makes sense that they should boost performance but it is reassurring to see that this is indeed consistently the case.
>
> On a more general level, higher order GNNs, and Schur nets are really not just a specific architecture but a class of
> architectures -- in particular the theory leaves open the question of how the subgraphs are selected and which subgraphs should
> communicate with which others. Implicit in your question is whether these choices should change when we
> move to different types of datasets or we move to a different task, e.g. link prediction. This is a very valid question, but we think it cannot be answered in just one paper. It will take the community some time to fully explore the vast design space of higher order networks. One thing that we are specifically working on is finding out whether in chemistry datasets including subgraphs corresponding to actual functional groups is helpful or not.
>
> COMMENTS. Thank you for appreciating that this paper is not just about a specific design variation on GNNs,
> but a general connection between spectral graph theory and automorphism-group equivariance, which, to the best of our knowledge, has not appeared in the literature previously. Automorphism group equivariance is important, but there have been few papers that explicitly exploited it in GNNs. Our hope is that the theory outlined in this paper will make it a little less daunting to
> incorporate automorpshism group equivariance in GNNs.

---

### Official Review · Reviewer_uFzV · 2024-07-08

**Soundness:** 2
**Presentation:** 1
**Contribution:** 3
**Rating:** 4
**Confidence:** 3

**Summary:**

This paper introduces Schur Net, an architecture designed to attain subgraph equivariance without fully determining the automorphism group. Utilizing spectral graph theory, Schur layers incorporate equivariant side-information from local structures to improve expressiveness. The authors have confirmed Schur Net's effectiveness through experiments conducted on the ZINC dataset.

**Strengths:**

- Efficiency. The proposed Schur Nets attain subgraph equivariance without the necessity to identify the full automorphism group of the subgraph, rendering it more efficient than the group theoretic approach.
- Theory. The authors have demonstrated the first-order permutation equivariance of Schur layers and have extended this to include higher-order permutation equivariance.
- Experiments. Schur Nets have shown impressive performance on the ZINC dataset.

**Weaknesses:**

- Presentation. This paper suffers from numerous spelling and grammar errors, as well as incorrect citations. The authors are advised to carefully proofread their work to amend these mistakes. Additionally, the theoretical background part is challenging to comprehend. The authors should work to simplify this part, incorporating only the necessary parts for the main results in the text and elaborating details in the appendix.
- Theory. The authors have indicated the use of only first-order activations; thus, it appears that only Corollary 1 is applied as the form of Schur Nets in the experiments. If this is the case, including Theorem 3 in the main text is unnecessary. The outcomes of Corollary 1 seem trivial, offering limited contribution.
- Expressiveness. Schur layers, as the paper admits, cannot express all equivariant linear maps, which constrains their expressiveness. The conclusion asserts that their method is "almost as expressive" as the group theoretic approach without providing supporting theorems. Furthermore, the paper lacks a thorough discussion on the precise expressiveness of Schur Nets and how they compare to other models.
- Missing Related Works. The absence of a related works section hinders the reader's ability to compare this study with prior research. The authors should introduce a section on related works and highlight the novelty of their study in comparison.
- Experiments. The scope of the experiments is too narrow, being limited to the ZINC dataset without a robust set of baseline models for comparison.

Despite the paper's subpar presentation, I acknowledge that my understanding may be flawed, and I am receptive to the authors' clarifications. Nevertheless, I urge the authors to enhance the paper's clarity and accessibility.

**Questions:**

How were Schur layers implemented in the experiments? Was equation (8) simply applied to the features of the subgraphs?

**Limitations:**

The limitations are outlined in Section 6, where the authors recognize the limited expressiveness of Schur Nets, the narrow range of experiments conducted, and the use of first-order activations.

---

> ### Author Rebuttal · Authors · 2024-08-06
>
> Thank you for your review and several suggestions that are very much on point.
>
> QUESTION 1. Yes, in our experiments we used for order Schur layers, which just use equation 8. What we like about our approach is that it bypasses all the group theory and reduces to something so simple. The rest of the network however, which
> involves message passing between different subgraphs is technically quite complicated because
> it has to account for how many vertices any pair of subgraphs have in common etc., as it is
> generally the case in higher order GNNs. In the main body of the paper we decided to focus only on Schur layers, because that is the novelty relative to other higher order message passing papers.
>
> Maybe we should emphasize that what gives our architecture its power is the combination of Schur layers with higher order message passing. Schur layers on their own, if they could only communicate via scalars, would not give very good performance. Conversely, higher order message passing on its own has already been shown to give state-of-the-art results, but we show (see also our more recent experiments mentioned in the global rebuttal) that take the automorphism group into account with Schur layers consistenltly improves performance further.
>
> FURTHER COMMENTS
>
> Thank you for pointing out the need to proofread the paper and increase the clarity of some sections. It
> is also a good idea to add a "Related works" section, the reason we decided to cite the literature inline
> was mostly to save space, but if the paper gets accepted, the extra page available in the camera ready
> version would allow us to add a separate section for this purpose.
>
> - Theory. While equation 8 is simple, we beg to differ about it being trivial. Since the early days of
> GNNs, researchers have proposed building GNNs based on the "graph Fourier transform", i.e., some kind
> of expansions of functions in the eigenvectors of the Laplacian. However these architectures were usually
> motivated by an analogy with convolution in Euclidean space. Somewhat independently, people studied
> GNNs from the point of view of equivariance alone and proposed corresponding architectures.
>
> Corollary 1 and Theorem 3 bring these two threads of work together, and show that equivariance to the
> automorphism group (of subgraphs), which seems like a complicated algebraic issue, can in fact be enforced
> via the Laplacian. The proof not only shows that this approach is equivariant but also why it is
> equivariant: because the eigenspaces of the Laplacian MUST transform independently under permutations.
> The only case that the spectral approach is weaker than a fully group theoretical approach (which to our
> knowledge has never been implemented) is when some of the eigenespaces further split into smaller
> invariant subspaces, i.e., if some of the eigenvalues of the Laplacian "accidentally" coincide.
> This gives a whole new interpretation to spectral GNNs and suggests that the reason they could achive
> relatively good performance was maybe not so much the analogy with convolution but simply that they
> could take the automorphism of the group into account. Of course the actual setting in our case is different
> because we apply Schur layers at the subgraph level rather than at the level of the whole graph.
>
> In summary, the reason that we included the theoretical results, including the proofs in the main body of the paper in full detail is that we think they are of independent importance. The proofs themselves shed light on a new connection between spectral graph theory and equivariance. Potentially this could be exploited in other settings as well. Further, it can be the basis of theoretical studies of expressivity (see below).
>
> - Limitations of expressiveness. We mainly mention the fact that Schur layers cannot necessarily
> account for all equivariant linear maps because we think that studying in what cases they can or cannot
> is an interesting theoretical research question. As mentioned above the "gap" only arises if some of the
> eigenvalues of the Laplacian coincide, while the corresponding subspaces are not, in fact, irreducible invariant
> spaces of the action of the automorphism group. We know from the literature on e.g. graph isomorphism that
> in the typical case this does not happen: unless there is some specific symmetry involved, the eigenvalues
> of moderate to large graphs are typically distinct. In fact, for most graphs they are sufficient to
> distinguish between non-isomorphic graphs. So studying this gap comes down to studying the "special"
> graphs that defeat the Schur layers and possibly there is an interesting connection to graph isomorphism.
> However, we felt that this would go beyond the scope of the present paper.
>
> - Regarding further experiments please see the "global rebuttal".

---

> > ### Comment · Reviewer_uFzV · 2024-08-11
> >
> > I appreciate the detailed rebuttal; however, my concerns regarding the paper persist. Primarily, the paper's presentation requires refinement. The theoretical sections (Section 3 and the initial part of Section 4) are hard to follow, and their connection to the proposed Schur layer is ambiguous. I recommend that the authors condense these sections to include only essential results. Additionally, the paper's theoretical contributions appear limited. Lemma 1, which seems rather obvious, is essentially Lemma 3.1 from [1], and Corollary 1 is a direct consequence of Lemma 1. Although Theorem 3 extends this to higher-order equivariance, it lacks experimental validation. I would also advise the authors to perform more experiments on larger datasets beyond the TU datasets. For these reasons, I am inclined to retain my original score.
> >
> > [1] Babai, L., Grigoryev, D. Y., & Mount, D. M. (1982, May). Isomorphism of graphs with bounded eigenvalue multiplicity. In Proceedings of the fourteenth annual ACM symposium on Theory of computing (pp. 310-324)..

---

> > > ### Author Response · Authors · 2024-08-13
> > >
> > > Thank you for your response! For experimental validation on other (relatively large) datasets, please
> > > see our result on OGBG-MOLHIV (81.6+-0.295) as reported in the global rebuttal and the detailed results
> > > table posted to Reviewer EFuW. Further comments:
> > >
> > > 1. Thank you so much for bringing the [Babai et al., 1982] paper to our attention. Lemma 1 by itself is
> > > indeed almost trivial (that's why we called it a Lemma not a Theorem). However, the [Babai et al.] paper is still an important reference because it underlines the potential connection to the literature on graph isomorphism. Specifically,
> > > there might be results in that paper or follow-up publications that allow us to put a bound on the potential
> > > expressivity gap between the group theoretical and the spectral approaches. We will look into that and
> > > potentially add a discussion to the Appendix. Thank you!!
> > >
> > > 2. The intended message of our paper is that something which other papers
> > > attempted to do in a complicated way with irreducible representations etc., can be done (almost) equally
> > > well in a simple way using just the eigendecomposition. From this point of view, in our opinion, the fact that Lemma 1
> > > and Corollary 1 are technically simple is more of a strength rather than a weakness.
> > >
> > > We do appreciate your point about presentation, however. We certainly didn't mean to make the description of the group theoretic approach look more complicated than it needs to be, so we will will rework Sections 3 and 4 to make them as simple and readable as possible. We appreciate that for readers who are not already familiar with the group theoretic approach,
> > > Section 3 might be too dense, so we will add more background information in the Appendix
> > > and remove unnecessary details from the main body of the paper. We will also add a description of how Linamps (from [Maron et al., 2019]) work, and provide illustrative examples.
> > >
> > > 3. Regarding the gap in expressiveness mentioned in your original review, please see the table in our new
> > > global comment comparing the number of invariant spaces produced by the group theoretical vs the spectral approach.
> > > We found that for the specific subgraphs we considered there is, in fact, no gap.
> > > actually there is no gap.

---

### Official Review · Reviewer_EFuW · 2024-07-16

**Soundness:** 4
**Presentation:** 3
**Contribution:** 3
**Rating:** 7
**Confidence:** 4

**Summary:**

This paper proposes to define permutation equivariant functions on subgraphs via spectral theory as opposed to the more traditional 'equivalence classes' of permutations [Maron et al.] The paper lays out the theory behind producing equivariant maps via the eigen decomposition of the subgraph's Laplacian and the permutation invariance of the eigenspaces, which along with standard arguments from representation theory produces equivariant functions, although not all of them. The validity and robustness of the approach is further exemplified in the experiments on benchmark tasks and datasets, comparing this approach to [Maron et al.]

[Maron et al.] Maron et al., Invariant and Equivariant Graph Networks NeurIPS 2019

**Strengths:**

1. This paper proposes a novel method that can significantly reduce the exponential search space of permutation equivariant layers between overlapping subgraphs in the P-tensor formalism.
2. This can be extended to Subgraph GNNs and other high order networks
3. Some experimental verification of good performance

**Weaknesses:**

1. The background is not clearly laid out, especially that of the P-tensor formalism. Without referencing back to these papers it wasn't possible for me to understand this manuscript. I suggest to add background in the supplementary material.

2. Other than the spectral convolution being permutation equivariant, there are no theoretical guarantees on its expressivity (in terms of Weisfeiler Leman or Subgraph GNNs.)

**Questions:**

1. I read the background but I still don't understand what this means in line 337, "In table 1, we see that Schur layer outperforms Linmaps under different cycles sizes, especially when only cycle 5 and 6 are considered. " Do you choose as your neurons as only cycles of certain size and then computing the equivariant layers in your proposed spectral approach?

---

> ### Author Rebuttal · Authors · 2024-08-06
>
> Thank you for your thoughtful comments and questions.
>
> QUESTION 1. We experimented with various architectures, but in the final experiments we just used subgraph-layers corresponding
> to cycles, edges and vertices. The cycles were of size 5 and 6 corresponding to the aromatic rings
> that commonly occur in molecules. The experiments show that applying the Schur-layer idea to just the cycles alone
> consistently improves performance over the baseline P-tensors model. The P-tensor model is the natural baseline here
> here since it is currently the state-of-the art model amongst subgraph-based higher order GNNs. The experiments (both those that are in the original paper and those that we have done since) show that automorphism-group aware convolutions with Schur layers within subgraphs robustly improve performance over the more traditional local convolutions that ignore the automorphism group of the subgraphs.
>
> QUESTION 2. There might be a slight misunderstanding here. The naive approach would not involve an exponential
> number of layers. It would involve first finding the automorphism group of each subgraph (which is a
> complicated combinatorial problem but could be done in precomputation) then symmetrizing over the
> automorphism group in each layer either by (a) enumerating each of its elements or (b) using the representation theory of
> the specific group. Technically, both finding the automorphism group and enumerating over its elements is
> NP-hard in the size of the subgraph. The real problem is not just the computational cost but how
> complicated considering all potential automorphism groups, generating the irreducible representations, etc.
> would be. We are not aware of any higher order GNN paper that seriously proposed doing this. Therefore, we
> cannot directly compare the running time of our model with this. What our paper shows is that all of
> this can be bypassed with a simple spectral graph theory "trick".
>
> A huge benefit of our approach is its flexibility. We can easily change to other subgraphs just by changing
> the subgraph template, the Laplacian etc are computed automatically, and there is no need to search for
> the automorphism group, somehow derive its irreducible representations, and so on, neither there is a need to
> sum over all group elements, which would have $O(m!)$ complexity in principle, where $m$ is the size of the
> subgraph.
>
> In practical terms, as we also explain in our response to Reviewer fyBT, the biggest computational bottleneck
> in higher order GNNs is orchestrating the message passing process in parallel across all subgraphs in a given
> layer, given that the subgraphs overlap with each other in different ways. To facilitate this we had to write
> a separate library complete with specialized CUDA kernels and a considerable amount of engineering.
> In practice, using this library, first order message passing, like in our experiments, is only 3-4 times
> slower than conventional GNNs. The actual Schur layers only make up a small fraction of the run time.
> If we summed over all elements of the automorphism group, however, we estimate that the automorphism-group equivariant
> layers would be about 100 times slower, but for the reasons cited above such an approach would be very clunky
> to implement anyway.
>
> QUESTION 3. There might be another slight point of confusion here. First of all, "Linmaps" is not the same as [Maron et al]'s original model, since all the layers in [Maron et al.]'s model operate on the entire graph, whereas we consider subgraph based models where the Linmaps happen at the level of subgraphs. The reason that [Maron et al.] is cited is that we apply the same linear transformations at each subgraph as they did at the level of the entire graph and they were the first to enumerate all such possible linear transformations.
>
> Secondly, the problem with "Linmaps" (as implemented in for example P-tensors) is not that they are expensive but that
> they are not expressive enough because they fail to take into account the automorphism group of the subgraph. Schur layers are a plug-in replacement for Linmaps that can take into account the automorphism group, which is basically a form of equivariant side information (see Sections 3 and 4). Fortunately, (see response to fyBT), using to the spectral graph theory trick, the added computational cost is minimal and in practice pales in comparison to the cost of using higher order message passing in the first place (please see the response to fyBT for the theoretical complexity). In summary, we have a more expressive model and achieve a consistent improvement over other higher order models like P-tensors without any appreciable extra computational cost.
>
> The reason we did not manage to get results on the full ZINC dataset is primarily a memory issue.
> As we explain, to achieve speed, we need to cache a variety of control data structures on the GPU and on the
> full ZINC dataset we simply ran out of memory. We are presently working on removing this limitation from the software by improving the engineering of the backend.
>
> FURTHER COMMENTS.
>
> - We appreciate your comment about the paper not being self-contained because it doesn't explain the mechanics of higher order message passing, especially since this is a relatively new formalism. In the main paper we just wanted to concentrate on what is novel, which is the Schur layer. We will add background information on P-tensors, etc. to the appendix.
>
> - See the global rebuttal for further results on the performance boost of Schur layers, unfortunately these only came after the deadline.
>
> - Thank you for bringing [Feng et al.] to our attention, we were not aware of it and the results on ZINC 12K are very impressive! One fundamental difference is that theirs is a fundamentally 2nd order model. While we have worked out the theory of higher order Schur layers we have not used them in our experiments. This paper gives a strong motivation to explore 2nd order Schur layers.

---

> > ### Comment · Reviewer_EFuW · 2024-08-08
> > **Response**
> >
> > I highly appreciate your detailed rebuttal.
> >
> > 1. What I'm missing is that if you only restrict to automorphisms of cycles and other simple subgraphs, isn't it computationally feasible to know the decomposition into irreducible representations? Isn't it the decomposition of the representation of the cyclic group acting on the vectors on the subgraph (`neuron')?
> >
> > 2. Thank you for the clarification. Yes, upon looking it over again the number of linear equivariant mappings you have is always larger as the automorphism group of the subgraph is a subgroup of $S_m$, where $m$ is the subgraph size.
> >
> > 3. "cache a variety of control data structures on the GPU": By this you mean the eigendecompositions of the subgraphs?
> >
> > 4. I agree with the comment re [Feng et al.].
> >
> >
> > I'm still debating whether this merits a score increase. is it possible to put together all the experimental results in a more readable format like a pdf? Or just a reply that is succinct with highlights from the best experimental claims you think from both the paper and rebuttal.
> >
> > Thank you

---

> > > ### Author Response · Authors · 2024-08-08
> > >
> > > Thanks so much for the quick response to our rebuttal.
> > >
> > > 1. For cycles the relevant group is not the cyclic group but the dihedral group, which is a little bit more complicated because it also has some two dimensional irreducible representations.  Table 4 shows that adding cycles with {1,2,3} branches hanging off them can improve performance, and here the automorphism group is different (see the paragraph titled "Flexibility" in the Experiments section). Without the Schur layer formalism we would have to work out the irreducible representations of all of these group separately by hand. However, for the "production run" used to generate the results of Table 5, we ended up just using cycles, so you make a fair point.
> > >
> > >  Our primary goal with the paper was to develop the general machinery for automorphism group aware message passing in higher order subgraph neural networks. then. After we have done that and wrote the corresponding software, we started working on validating the approach, trying to show that adding automorphism-group aware operations can improve the state-of-the-art in empirical results on benchmark datasets. What we had in mind was adding subgraphs corresponding to actual functional groups.
> > >
> > >   We found that just by adding cycles we can already beat the other papers on ZINC 12K (with the exception of [Feng et al.] which you kindly pointed out but we didn't know about). In addition to branched cycles, we also did some experiments on star-shaped subgraphs for example, but in terms of producing results for this conference deadline we just concentrated on cycles in the end because that seemed like the most direct path to competing with the other algorithms. Our longer term goal is to explore much more adventurous applications of this framework.
> > >
> > >   In summary, you are right, if all that we were interested in were cycles and first order message passing, then we could have developed an architecture specialized to just that and that would have been easier maybe. It would probably be a rather incremental contribution to the field, though.
> > >
> > > 3. We wish it were just the eigenvectors that need to be cached. The main technical difficulty in GNNs is parallelizing the message passing step on the GPU, given that the graph structure is not regular. Most of the community uses PyTorch Geometric, which solves this problem for classical message passing GNNs by reducing the message passing step to one big "scatter" operation. The problem with using this for higher order message passing is that in architectures like ours for a given pair of sending and receiving subgraphs, the exact form of the message passing map between them depends on how many vertices they have in common and which vertices those are. So effectively the message passing operation is different for each pair of (sending, receiving) subgraphs.
> > >
> > >   To solve this problem and be able train our model at a comparable speed to other GNNs, we had to write specialized CUDA kernels and ultimately a separate library. The kernels take as input the source subgraph-layer, a data structure specifying which subgraph communicates with which subgraph (in most of our experiments this is just determined by whether they overlap or not), as well a data structure that specifies for each pair of (source,destination) subgraphs which vertices they share. The latter two data structures are relatively expensive to compute because they need to be created on the CPU so it makes sense to cache them (on the GPU). This is what can lead to memory issues. It also complicates the batching process, because unlike in PyTorch Geometric, we can't just batch the graphs by merging them into one big graph. We are working on relieving the memory issue by updating the library so that it can move these data structures in and out of GPU memory flexibly, but using a preloading strategy so as to hide the latency of the memcpy calls. The overall goal is that the library should make higher order GNNs just as easy to use as regular message passing networks, hiding all these details on the backend. We should also not that the library can also do second order message passing as described in Theorem 3, we just haven't had a chance to experiment with that yet.
> > >
> > > ----
> > >
> > > I don't think we can submit pdf's at this stage, but we will compile the additional results in table format and share them soon.

---

> > > > ### Author Response · Authors · 2024-08-09
> > > > **updated results table**
> > > >
> > > > Please find the updated Table 5 below:
> > > >
> > > > | Model                          | ZINC-12K MAE (↓) | OGBG-MOLHIV ROC-AUC(% ↑) |
> > > > |--------------------------------|------------------|--------------------------|
> > > > | GCN [Kipf and Welling, 2017]    | 0.321 ± 0.009    | 76.07 ± 0.97             |
> > > > | GIN [Xu et al., 2019]           | 0.408 ± 0.008    | 75.58 ± 1.40             |
> > > > | GINE [Hu et al., 2020]          | 0.252 ± 0.014    | 75.58 ± 1.40             |
> > > > | PNA [Corso et al., 2020]        | 0.133 ± 0.011    | 79.05 ± 1.32             |
> > > > | HIMP [Fey et al., 2020]         | 0.151 ± 0.002    | 78.80 ± 0.82             |
> > > > | CIN [Bodnar et al., 2021]       | 0.079 ± 0.006    | 80.94 ± 0.57         |
> > > > | DS-GNN (EGO+) [Bevilacqua et al., 2022] | 0.105 ± 0.003    | 77.40 ± 2.19             |
> > > > | DSS-GNN (EGO+) [Bevilacqua et al., 2022] | 0.097 ± 0.006    | 76.78 ± 1.66             |
> > > > | GNN-AK+ [Zhao et al., 2022]     | 0.091 ± 0.011    | 79.61 ± 1.19             |
> > > > | SUN (EGO+) [Frasca et al., 2022] | 0.084 ± 0.002    | 80.03 ± 0.55             |
> > > > | First order P-tensors  [Hands et al., 2024] | 0.071 ± 0.004 | 80.76 ± 0.82         |
> > > > | **Schur-Net (ours)**            | **0.064 ± 0.002** | **81.6 ± 0.295**         |

---

> > > > > ### Comment · Reviewer_EFuW · 2024-08-12
> > > > > **Concerns addresses; score raised**
> > > > >
> > > > > Overall, using the eigen decomposition to produce equivariant functions is novel and is a standard operation for GNNs. The empirical results are non-marginal. Score raised to 7.

---

### Author Rebuttal · Authors · 2024-08-07

We thank all the reviewers for their careful reading of our paper and thoughtful comments. Points raised by individual reviewers are addressed in the individual responses, here we would like to make some general comments.

- First of all, we stress that our paper has two separate aims: 1. Making a general theoretical contribution to the literature by pointing out a connection between spectral graph theory and the theory of equivariant GNNs, specifically, how the eigendecomposition of the graph Laplacian can "mimick" the decomposition into irreducible subspaces that is at the heart of the mathematically more sophisticated group theory based approach to equivariance. 2. Making a practical contribution by showing how the theoretical results can be used to easily "upgrade" higher order subgraph-based GNN to take into account the automorphism group of subgraphs as  "side information". This upgrade makes the GNNs strictly more expressive at little additional computational cost.

- Several reviewers ask about the potential gap in expressivity related to the fact that in principle the Laplacian-based decomposition into invariant subspaces might be coarser than the group theory based approach. We explain that finding the cases in which this happens is quite a deep theoretical question that goes beyond the scope of the present paper. In practical cases for moderate sized subgraphs however we argue that there is little or no gap.

- Regarding the comments about the limited scope of the experiments, we have now conducted further experiments, and found that they confirm that taking the automorphism group of subgraphs into account with Schur layers boosts the performance of higher order subgraph neural networks. For example, on the classic TUDatasets we find:

  | Dataset   | *Linmaps*   | SchurLayer  |
  |-----------|-----------|-------------|
  | Proteins  | 74.7±3.8  | **75.4±4.8**|
  | MUTAG   |   89.9±5.5  | **90.9±4.7**   |
  | PTC\_MR   | 61.1±6.9  | **64.6±5.9**|
  | NCI1      | 82.1±1.8  | **82.7±1.9**|

  Importantly, on the OGBG-MOLHIV dataset we achieve an ROC-AOC of 81.6%+- 0.295, whereas a corresponding P-tensor model without Schur-layers only achieves 77.925% +- 2.461. Please note that this result is highly significant on its own terms, in particular it beats all other competing models (GINE, PNA, HIMP, CIN, SUN, etc.) cited in the P-tensor paper [Hands et al, 2024].

- Several reviewers also ask the computational cost of our model. In addition to our detailed comments about computational complexity and the practical challenges of implementing higher order message passing, here are some illustrative single GPU per-epoch wall-clock runtimes:

  | Dataset   | Linmaps | SchurLayer |
  |-----------|---------|------------|
  | ZINC-12k  | 25.4s   | 27.6s      |
  | NCI1      | 9.5s    | 11.5s      |

  **Table 2**: Runtime per epoch with hyper-params num_layers = 4, rep_dim = 128, dropout = 0.0, batch_size = 256, cycle_sizes = 3,4,5,6,7,8. The implementation of Linmaps and SchurLayer is based on our internal software.

---

> ### Author Response · Authors · 2024-08-13
> **number of irreducible spaces**
>
> We thank all the reviewers for their comments and the discussion that followed. Since several of you asked about the potential expressivity gap between the group theoretic and spectral approach, we decided to investigate this empirically for the subgraphs that we used (or considered using) in our experiments. The table below compares the number of invariant subspaces found using the two approaches for the first order case. The closer the number of spectral subspaces is to the number of subspaces found by the algebraic approach (which is the theoretical upper limit), the more expressive the Schur layers are. (The number of irreducible subspaces is computed by finding all irreps that occur in the action of $\textrm{Aut}_G$ on $\mathbb{R}^m$ and adding their multiplicities. This is easiest to do with character theory.)
>
>
> | Graph                                       | Automorphism group | # of distinct eigenvalues (# of independent linear maps found by SchurLayer) | # of irreducible subspaces (# of linear maps found by group theoretic approach) |
> |---------------------------------------------|--------------------|-----------------------------------------------------------------|-----------------------------------------------------------------|
> | 6-cycle                                     | $D_6$                | 4                                                               | 4                                                               |
> | 5-cycle                                     | $D_5$                | 3                                                               | 3                                                               |
> | 4-cycle                                     | $D_4$                | 3                                                               | 3                                                               |
> | 3-cycle                                     | $D_3$                | 2                                                               | 2                                                               |
> | 5-star                                      | $S_4$                | 3                                                               | 3                                                               |
> | 4-star                                      | $ S_3 $               | 3                                                               | 3                                                               |
> | 3-path                                      | $S_2$                | 3                                                               | 3                                                               |
> | n-cliques                                   | $ S_n $               | 2                                                               | 2                                                               |
> | 5-cycle with one branch                     |  $S_2$                | 6                                                               | 6                                                               |
>
> Remarkably, we see that there is in fact no gap between the algebraic and the spectral approach! We are very intrigued to find some class of subgraphs for which there is a gap and to characterize that class. We are currently working on this topic.
>
> Please note that for the baseline case of full $S_m$ equivariance there would only be two linear maps in each case. Also note that due to the way that product representations work, when there is no gap for a given subgraph in the first order case, there would be no gap in higher order cases either.
>
> Once again, thank you for your careful reading of our paper and all your suggestions on how to improve it!

---

### Decision · Program_Chairs · 2024-09-25

**Decision:**

Accept (poster)

**Comment:**

This paper brings forth a new method based on spectral graph theory to achieve permutation equivariance in subgraph-based graph neural networks (GNNs). By constructing a basis for equivariant operations directly from the graph Laplacian, the paper bypasses the need to explicitly determine automorphism groups, which is computationally expensive. This approach helps to enhance the expressiveness of GNNs when dealing with complex local structures, such as cycles within molecules/subgraphs.

The reviewers recognized the significance of this contribution, noting that Schur Nets offer an efficient and theoretically sound method for incorporating equivariant operations in GNNs. The additional experiments added during the rebuttal phase, particularly on the MOLHIV dataset, provided further empirical validation of the method's effectiveness, addressing some concerns about the initial limited scope of evaluation.

On the flip side, issues with the clarity of the theoretical sections were consistently highlighted, with some reviewers finding the explanations of key parts of the paper difficult to follow. While the additional experiments help to mitigate some concerns, further refinement of the paper's presentation, particularly in simplifying and better explaining the theoretical contributions, would strengthen the work.

Given the novelty of the approach, the empirical improvements demonstrated, and the potential impact on the field, I recommend accepting this paper, with the expectation that the authors will continue to refine the clarity and depth of their explanations in the final version.